# Cadherins orchestrate specific patterns of perisomatic inhibition onto distinct pyramidal cell populations

Julie Jézéquel [1,2], Giuseppe Condomitti [1,2], Tim Kroon [1,2], Fursham Hamid [1,2], Stella Sanalidou [1,2], Teresa Garcés[1,2], Patricia Maeso [1,2], Maddalena Balia[1,2], Zhaohui Hu[1,2], Setsuko Sahara[1,2] & Beatriz Rico [1,2] ✉

GABAergic interneurons were thought to regulate excitatory networks by establishing unselective connections onto diverse pyramidal cell populations, but recent studies demonstrate the existence of a cell type-specific inhibitory connectome. How and when interneurons establish precise connectivity patterns among intermingled populations of excitatory neurons remains enigmatic. We explore the molecular mechanisms orchestrating the emergence of cell type-specific inhibition in the mouse cerebral cortex. We demonstrate that layer 5 intra- (L5 IT) and extra-telencephalic (L5 ET) neurons express unique transcriptional programs, allowing them to shape parvalbumin- (PV+) and cholecystokinin-positive (CCK+) interneuron wiring. We identified *Cdh12* and *Cdh13*, two cadherin superfamily members, as underpinnings of cell type- and input-specific inhibitory patterns of L5 pyramidal cell populations. Multiplex monosynaptic tracing revealed a minimal overlap between IT and ET presynaptic inhibitory networks and suggests that different PV+ basket cell populations innervate distinct L5 pyramidal cell types. Here, we unravel the contribution of cadherins in shaping cell-type-specific cortical interneuron wiring.

Precise and fine-tuned neuronal connectivity underlies even the most basic animal behavior. This connectivity reaches an extreme complexity in the mammalian cerebral cortex, where two main types of neurons coexist: glutamatergic principal or pyramidal cells and gamma-aminobutyric acid-expressing (GABAergic) interneurons. Pyramidal cells are highly heterogeneous and can be organized into distinct neuronal ensembles, forming unique information-processing streams[1,2]. By selectively targeting different subcellular compartments of pyramidal cells, inhibitory interneurons play a critical role in gating cortical activity and ensuring proper brain computation[3]. However, how interneurons regulate such intricate and diverse excitatory networks remains enigmatic.

The classical vision assumed that interneurons indiscriminately target all pyramidal cells in the cerebral cortex[4–6]. However, there is

now substantial evidence indicating that cell-type-specific connectivity motifs underlie interneuron wiring[7–13]. These studies suggest that interneurons discriminate among different postsynaptic targets and exhibit biased connectivity patterns[7,8,12,14–16]. For example, pyramidal cells projecting to subcortical areas receive more perisomatic inhibition from parvalbumin-expressing (PV+) basket cells than local-projecting neighboring cells[8,15]. Similarly, cholecystokinin-expressing (CCK+) basket cells preferentially target pyramidal cells sending axons outside the hippocampus[7]. Consequently, interneurons not only influence pyramidal cell activity by targeting different subcellular compartments but they can also gate the information flow of distinct subnetworks of pyramidal cells.

How interneurons select specific targets during development and what molecular programs control such refined connectivity motifs to

[1]Centre for Developmental Neurobiology, Institute of Psychiatry, Psychology and Neuroscience, King's College London, London, United Kingdom. [2]MRC Centre for Neurodevelopmental Disorders, King's College London, London, United Kingdom. ✉e-mail: beatriz.rico@kcl.ac.uk

arise is still unknown. Here, we took advantage of the organization of cortical layer 5 (L5) and the coexistence of two pyramidal cell types with distinct projection targets to explore the emergence and molecular determinants of cell-type-specific perisomatic inhibition. After confirming that L5 intra- (L5 IT) and extra-telencephalic (L5 ET) pyramidal cell populations[16–21] in the somatosensory cortex receive different ratios of PV+ and CCK+ inputs, we found that two members of the Cadherin superfamily, *Cdh12* and *Cdh13*, contribute to instructing cell-type- and input-specific perisomatic inhibition. Simultaneous mapping of the inhibitory connections targeting L5 IT and L5 ET neurons revealed the contribution of different populations of PV+ basket cells to distinct pyramidal cell populations. Altogether, our findings unveil a molecular and cellular code orchestrating the assembly of inhibitory microcircuits targeting different pyramidal cell populations.

## Results

### Differential perisomatic inhibition onto distinct pyramidal cell-types

Previous studies in the hippocampus, entorhinal cortex and prefrontal cortex have shown that the levels of perisomatic inhibition differ between pyramidal cells with distinct projection targets[7–11,13,15]. To investigate whether this principle is maintained across cortical areas, we took advantage of the stereotypical organization of layer L5 in the primary somatosensory cortex (S1), where distinct pyramidal cell types can be segregated based on their axonal projections[17]. To simultaneously label L5 IT and L5 ET pyramidal cells, we co-injected RB488 and RB555 fluorophore-coated latex retrobeads in the contralateral S1 and the ipsilateral pons of wild-type mouse neonates (Fig. 1a and Supplementary Fig. 1a). We chose cortico-pontine projecting neurons as a reference for the L5 ET population as early postnatal injections in the pons provided the most reliable labeling among the different subcortical areas tested. Retrograde tracing from the injection sites revealed a double-layer cell distribution whereby L5 ET-labeled neurons were restricted to L5b, and L5 IT-labeled cells were enriched in L5a but present across the entire layer (Fig. 1a). Whole-cell voltage-clamp recordings from neighboring L5 IT- and L5 ET-labeled neurons revealed similar miniature inhibitory postsynaptic currents (mIPSC) amplitudes in both populations, but L5 ET neurons displayed higher mIPSC frequency than L5 IT neurons (Fig. 1b). These results thus indicate that S1 L5 ET neurons receive significantly more inhibitory inputs than the L5 IT population, corroborating previous observations of cell-type specific perisomatic inhibition in other cortical areas.

### Cell-type-specific inhibitory motifs onto pyramidal cell populations

PV+ and CCK+ basket cells are the two major sources of perisomatic inhibition in the cerebral cortex[22,23]. To investigate the density of PV+ and CCK+ synapses onto the soma of L5 IT and L5 ET neurons at P30, we used the specific presynaptic markers Synaptotagmin 2 (SYT2+)[24] and Cannabinoid Receptor 1 (CB1R)[25] to label PV+ and CCK+ inputs, respectively. We found that PV+ boutons were more abundant on L5 ET neurons than on L5 IT cells (Fig. 1c, d), irrespective of their position in L5a or L5b (Supplementary Fig. 1b). In contrast, L5 IT neurons located in L5a and L5b received more CCK+ boutons than L5 ET cells (Fig. 1f, g and Supplementary Fig. 1b) although we observed a gradual decrease of CCK+ innervation from L5a towards L5b (Supplementary Fig. 1b, d). It is also important to note that CCK+ inhibition remained largely outnumbered by PV+ inhibition in both populations (Supplementary Fig. 1c, d), suggesting that our mIPSC recordings likely reflect PV+ innervation (Fig. 1b).

To investigate when these distinct patterns of perisomatic inhibition emerge, we next characterized the developmental timeline of both basket cell inputs onto L5 IT and L5 ET neurons. We observed that the differences in PV+ innervation between both L5 populations were already present at the early stages of synapse formation (P10) and

further increased throughout postnatal development (Fig. 1e). CCK+ inhibition followed a slower progression and got gradually more robust over time (Fig. 1h). Altogether, our results indicate that cell type-specific inhibition patterns onto distinct L5 pyramidal cell populations arise during synaptogenesis and independently of the target cell's location, which suggests that these precise connectivity patterns might represent intrinsic features of L5 pyramidal cell identity.

### Cell-type-specific transcriptional programs in distinct pyramidal cells

We then examined the transcriptomic signatures of L5 IT and L5 ET neurons to explore the molecular mechanisms underlying L5 cell type-specific perisomatic inhibition. We hypothesized that cell-type-specific synaptic programs allow L5 IT and L5 ET populations to shape interneuron connectivity in order to match their inhibitory needs. To investigate whether postsynaptic molecules could instruct differential inhibition, we dissected S1 deep layers at P10 - when the differential perisomatic inhibition patterns first emerged - and isolated retrogradely labeled L5 IT and L5 ET neurons using fluorescence-activated cell sorting (Fig. 2a and Supplementary Fig. 2). We then performed bulk RNA-sequencing (RNAseq) and compared the gene expression profiles of both pyramidal cell populations (Fig. 2a and Supplementary Fig. 3a). Using this approach, we identified multiple cell type-specific genes, including known markers of L5 IT neurons such as *Lmo4* and *Satb2*, and of L5 ET cells like *Bcl11b* and *Crym* (Supplementary Fig. 3b). Astroglia or oligodendroglia genes were absent from our dataset, however, we unexpectedly found some microglial genes in the L5 ET population (Supplementary Fig. 3c). Although markers for L5 ET neurons were still enriched, this contamination was predicted to affect read sampling of low-expressing genes and its downstream differential analysis. To obtain a more precise "L5 ET signature", we combined our dataset with two reference studies that characterized L5 pyramidal cell identity during early postnatal development[26,27] (Fig. 2b and Supplementary Fig. 3d, also see Methods). We identified 853 DEGs based on the union of these 3 datasets (Fig. 2c and Supplementary Fig. 3e), which showed an enrichment of genes involved in the organization and function of the synapse as unveiled by our Gene Ontology (GO) analysis (Fig. 2d). L5 IT-enriched genes contributed to a higher proportion of these synaptic GO terms, but terms relevant to perisomatic inhibition, such as "GABAergic synapse" and "cell-cell adhesion" also featured in the analysis were equally represented in L5 IT and L5 ET cells (Supplementary Fig. 3f).

Several families of cell-surface molecules have been implicated in cell-cell type recognition during neural circuit development[28–33]. We thus curated the DEGs list for genes coding cell-surface molecules, with the underlying hypothesis that interactions mediated by cell-surface molecules differentially expressed by L5 pyramidal populations might play a pivotal role in shaping cell-type-specific CCK+ and PV+ basket cell connectivity. To do so, we established a list of DEGs coding for cell-surface molecules specific to each L5 pyramidal cell population, and sought matching partners (using a combination of several datasets[28–30,34], see Methods) expressed in presynaptic interneurons based on their 1) specificity and 2) level of expression in CCK+ versus PV+ basket cells (Fig. 2e). Two cadherin superfamily members – Cadherin-12 (*Cdh12*) for L5 IT and Cadherin-13 (*Cdh13*) for L5 ET - appeared as top candidate genes from our final ranking (Fig. 2f). To validate our bioinformatic predictions, we performed single-molecule RNA fluorescent in situ hybridization and measured *Cdh12* and *Cdh13* transcript levels at P10 in L5 pyramidal cell populations retrogradely labeled with adeno-associated viruses (AAVs) expressing fluorescent reporters (Supplementary Fig. 4a). *Cdh12* RNA expression was three times more enriched in L5 IT than L5 ET neurons (Supplementary Fig. 4b–d). Conversely, *Cdh13* RNA was twice more abundant in L5 ET than in L5 IT neurons (Supplementary Fig. 4e–g). Altogether, these

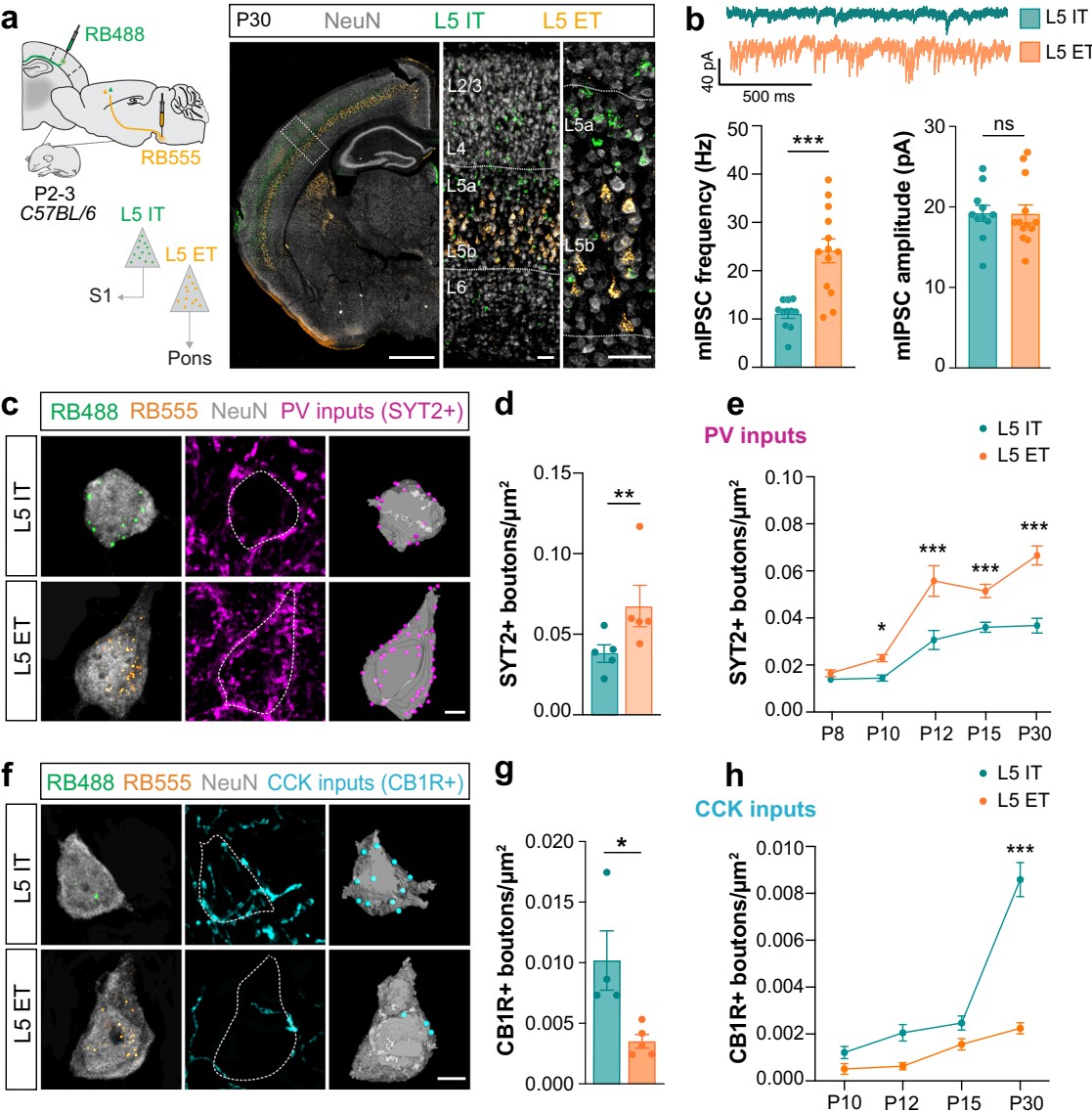

**Fig. 1 | Differential perisomatic inhibition onto distinct L5 pyramidal cell types.** **a** Retrograde labeling strategy. **b** mIPSCs frequency (Hz) (two-sided t-test *** $p = 0.0002$) and amplitude (pA) (two-sided paired t-test $p = 0.9563$) from retrogradely-labeled L5 IT ($n = 11$ cells, 5 mice) and L5 ET cells ($n = 13$ cells, 4 mice) at P30. **c, f** Confocal and 3D-reconstructed images illustrating PV (SYT2+) and CCK inputs (CB1R+) onto L5 IT and L5 ET somas. **d, g** SYT2+ (*L5 IT $n = 5$ mice, 62 cells; L5 ET $n = 5$ mice, 92 cells; two-sided paired t-test **$p < 0.01$) and CB1R+ (*L5 IT $n = 4$ mice, 82 cells; L5 ET $n = 5$ mice, 83 cells; two-sided t-test *$p = 0.0209$) boutons density at P30. **e** SYT2+ boutons density over post-natal development (Two-way ANOVA, Time factor ***$F_{(4, 1079)} = 46.89$, $p < 0.0001$, Cell type factor ***$F_{(1, 1079)} = 59.50$, $p < 0.0001$, Interaction ***$F_{(4, 1079)} = 5.025$, $p = 0.0005$; Tukey's posthoc: *P8* L5 $n = 5$ mice (*IT*, 60 cells, *ET*, 116 cells), $p = 0.6185$; *P10* L5 $n = 6$ mice (*IT*,

131 cells, *ET*, 123 cells), *$p = 0.0431$; *P12* L5 $n = 5$ mice (*IT*, 103 cells, *ET*, 109) cells), ***$p < 0.0001$; *P15* L5 $n = 7$ mice (*IT*, 164 cells, *ET*, 125 cells), ***$p = 0.0001$; *P30* L5 $n = 5$ mice (*IT*, 61 cells, *ET*, 97 cells), ***$p < 0.0001$). **h** CB1R+ boutons density over post-natal development (Two-way ANOVA, Time factor ***$F_{(3, 650)} = 44.86$, $p < 0.0001$, Cell type factor ***$F_{(1, 650)} = 43.81$, $p < 0.0001$, Interaction ***$F_{(3, 650)} = 20.51$, $p < 0.0001$; Tukey's posthoc: *P10* L5 $n = 5$ mice (*IT*, 73 cells, *ET*, 45 cells), $p = 0.3839$; *P12* L5 $n = 3$ mice (*IT*, 57 cells, *ET*, 49 cells), $p = 0.0834$; *P15* L5 $n = 5$ mice (*IT*, 107 cells, *ET* 80 cells), $p = 0.1494$; *P30* L5 $n = 5$ mice (*IT*, 127 cells, *ET*, 120 cells), ***$p < 0.0001$). In (**e, h**), each dot represents an average value of several mice. Data are represented as mean ± SEM, *$p < 0.05$; **$p < 0.01$; ***$p < 0.001$; ns not significant. Scale bars: 5 µm (**c, f**), 50 µm, 500 µm (**a**). Source data are provided as a Source Data file.

findings reveal that L5 IT and L5 ET express a unique cadherin signature, which mirrors cell-type-specific PV+ and CCK+ basket cell connectivity motifs onto L5 pyramidal cell populations.

## Specific inhibition of pyramidal cell populations requires *Cdh12* and *Cdh13*

To assess the functional role of postsynaptic cadherin expression, we first characterized CDH12 and CDH13 synaptic localization in L5 pyramidal neurons. To do so, we engineered Cre-dependent plasmids encoding hemagglutinin (HA)-tagged *Cdh12* or *Cdh13*, which were then electroporated into *Nex^Cre* mouse embryos at embryonic

day (E) 12.5 (Supplementary Fig. 5a). Overexpression of tagged-transmembrane proteins has been shown to fairly predict endogenous protein expression[35,36]. Both exogenous cadherins showed a punctate pattern around the soma of electroporated L5 pyramidal neurons (Supplementary Fig. 5b). Co-labeling of PV+ and CCK+ boutons revealed that postsynaptic CDH13-HA preferentially clustered near the postsynaptic sites of PV+ boutons, while CDH12-HA did not show any preferential localization near PV+ or CCK+ synapses (Supplementary Fig. 5c, d).

To investigate if CDH12 and CDH13 exert any synapse-specific function, we next performed cell-type-specific loss-of-function

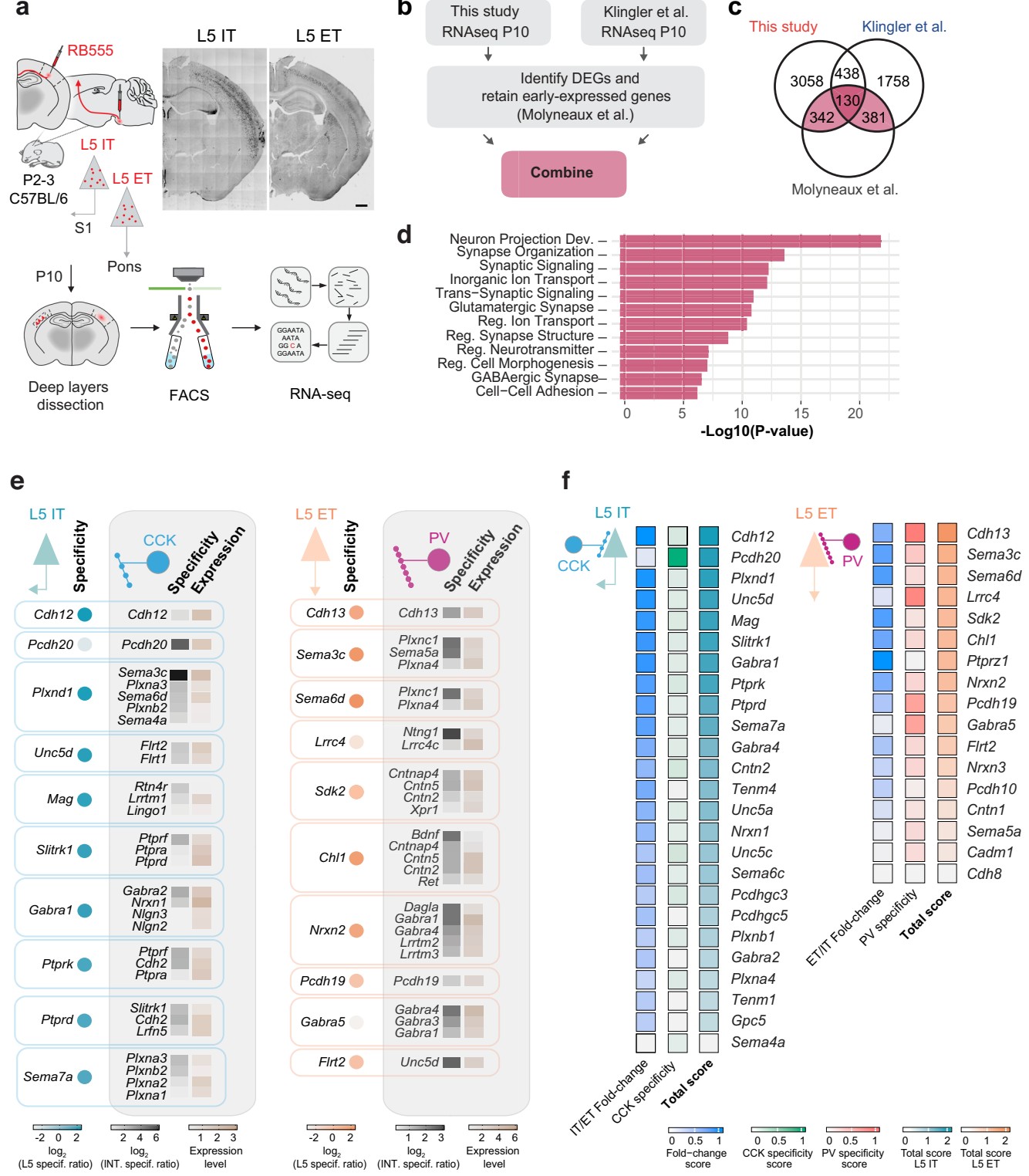

**Fig. 2 | Cell type-specific transcriptional programs in L5 pyramidal cell populations. a** Experimental design. **b** Combinatorial strategy of our RNAseq dataset with two other reference studies. **c** Number of differentially expressed genes (DEGs) between L5 IT and L5 ET neurons in the three RNAseq datasets. 853 DEGs (light and dark pink) were detected based on the union of the three datasets, and were used for downstream analysis. **d** Gene ontology terms significantly enriched in the final dataset. **e** Heatmaps showing cell-surface matching partner predictions between presynaptic basket cell interneurons and L5 pyramidal cell populations. DEGs coding for cell-surface molecules in L5 IT (blue dot) and L5 ET (orange dot) were ranked based on their enriched expression in a specific L5 population

(L5 specificity ratio). Their putative pre-synaptic partners were also ranked based on the specificity (gray square) and the level of their expression (brown square) in CCK+ versus PV+ basket cells (Interneuron specificity ratio). This analysis relies on known protein-protein interactions and gene expression in CCK+ and PV+ cell types described as such[80]. Note that the diagram only depicts one of the 2 components of inhibition. **f** Heatmaps showing the final selection criteria for L5 IT and L5 ET DEGs coding for cell-surface molecules. DEGs were ranked according to a total score, resulting from the L5 IT/ET fold-change expression and the CCK+/PV+ basket cell specificity score established in panel (**f**). Scale bar: 500 μm (**a**).

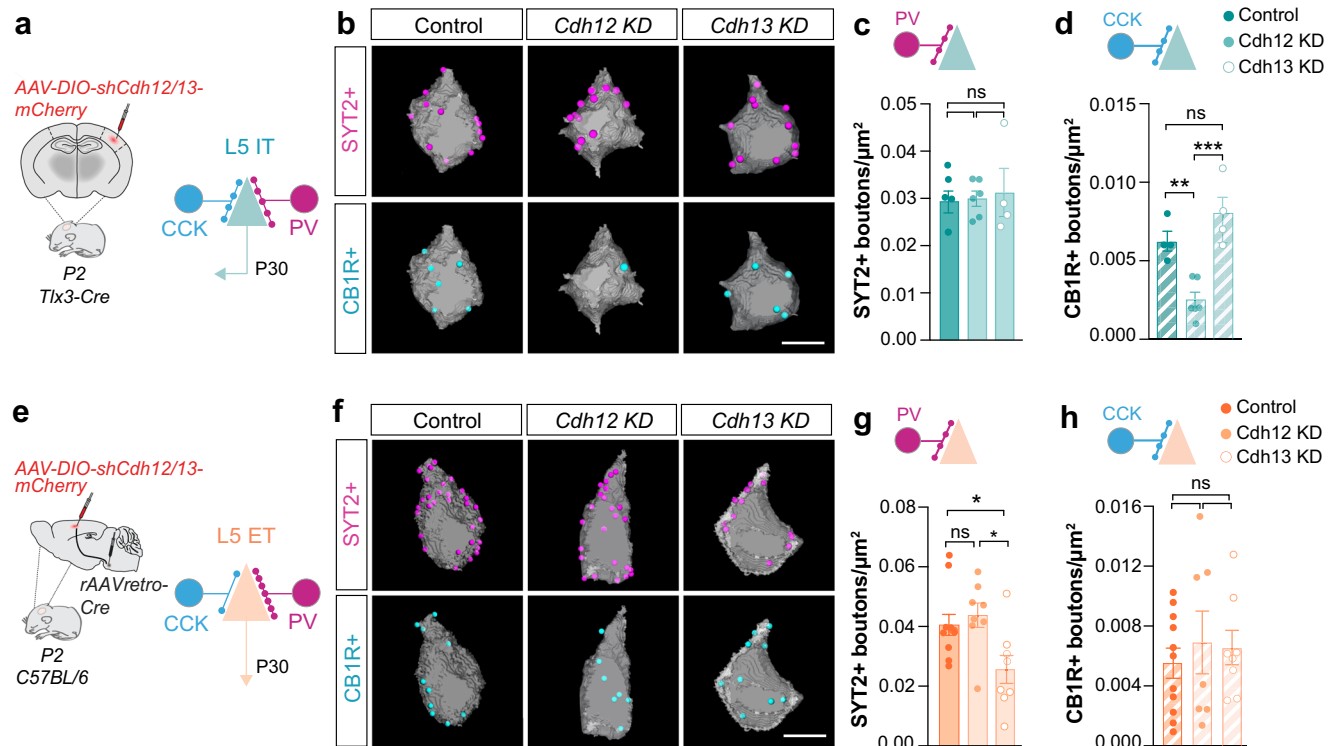

**Fig. 3 | Cadherins drive cell-type- and input-specific inhibition onto L5 pyramidal cell populations. a, e** Conditional viral strategy to down-regulate *Cdh12* and *Cdh13* in L5 IT and L5 ET neurons and measure its impact on perisomatic inputs at P30. **b, f** Representative 3D-reconstructed images of PV (SYT2+) and CCK inputs (CB1R+) onto L5 IT and L5 ET somas. **c** SYT2+ boutons density onto L5 IT (*Control*, *n* = 4 mice, 94 cells; *Cdh12 KD*, *n* = 6 mice, 140 cells; *Cdh13 KD*, *n* = 4 mice, 119 cells; One-way ANOVA *p* = 0.1136, Tukey's posthoc: Control vs *Cdh12* KD ns, Control vs *Cdh13* KD ns, *Cdh12* KD vs *Cdh13* KD ns). **d** CB1R+ boutons density onto L5 IT (*Control*, *n* = 4 mice, 95 cells; *Cdh12 KD*, *n* = 6 mice,141 cells; *Cdh13 KD*, *n* = 4 mice, 107 cells; One-way ANOVA ***p* = 0.0002, Tukey's posthoc: Control vs *Cdh12* KD

***p* = 0.0017, Control vs *Cdh13* KD *p* = 0.2888, *Cdh12* KD vs *Cdh13* KD ****p* = 0.0002). **g** SYT2+ boutons density onto L5 ET (*Control*, *n* = 11 mice, 291 cells; *Cdh12 KD*, *n* = 8 mice, 153 cells; *Cdh13 KD*, *n* = 8 mice, 170 cells; One-way ANOVA **p* = 0.0143, Tukey's posthoc: Control vs *Cdh12* KD *p* = 0.8291, Control vs *Cdh13* KD **p* = 0.0401, *Cdh12* KD vs *Cdh13* KD **p* = 0.0180). **h** CB1R+ boutons density onto L5 ET (*Control*, *n* = 11 mice, 220 cells; *Cdh12 KD*, *n* = 7 mice, 133 cells; *Cdh13 KD*, *n* = 8 mice, 141 cells; One-way ANOVA *p* = 0.2959, Tukey's posthoc: Control vs *Cdh12* KD ns, Control vs *Cdh13* KD ns, *Cdh12* KD vs *Cdh13* KD ns). Data are represented as mean ± SEM, **p* < 0.05; ***p* < 0.01; ****p* < 0.001; ns not significant. Scale bars: 10 μm (**b, f**). Source data are provided as a Source Data file.

experiments using a conditional knockdown (KD) strategy in vivo[37]. We designed Cre-dependent short-hairpin RNAs against *Cdh12* and *Cdh13* (*shCdh12*, *shCdh13*), which were compared to a control RNA (*shLacZ*)[37]. To specifically down-regulate the expression of our candidate genes in L5 IT neurons, we injected Cre-dependent *shRNA* AAVs in the S1 of *Tlx3-Cre* mice[38] (Fig. 3a). L5 ET neurons were targeted via co-injection of a retrograde Cre-expressing AAV (*rAAV2retro-Cre*) in the pons and the same *shRNA* AAVs in S1 (Fig. 3e). After validating the efficiency of our down-regulation approach in vitro and in vivo (Supplementary Figs. 6 and 7), we then quantified the impact of *Cdh12* and *Cdh13* KD on L5 IT and L5 ET perisomatic inhibition at P30. We found that *Cdh12* KD (32%) did not impact PV+ innervation but significantly reduced CCK+ inputs onto L5 IT cells (Fig. 3b–d). Conversely, *Cdh13* KD (22%) reduced the number of PV+ inputs targeting L5 ET neurons without affecting CCK+ inhibition (Fig. 3f–h). Consistent with these observations, mIPSCs frequency was reduced in L5 ET neurons expressing *Cdh13* KD and was concomitant with a decrease in mEPSC frequency and amplitude (Supplementary Fig. 8). Changes in mEPSC parameters could be due to compensatory mechanisms secondary to a reduced inhibition as previously described[39] although we cannot discard a potential role of *Cdh13* in instructing specific excitatory inputs onto L5 ET pyramidal cells. We did not observe any effect of *Cdh12* KD on the inhibitory and excitatory currents of L5 IT neurons (Supplementary Fig. 8), which may be due to the small

contribution of CCK+ inputs to the total perisomatic inhibition received by these cells (Supplementary Fig. 1b, c).

To uncover if this effect was conditioned by cell-type identity, we also assessed the impact of *Cdh12* KD in L5 ET neurons and *Cdh13* KD in L5 IT neurons. Despite displaying similar down-regulation levels (compare Supplementary Fig. 7b, d, f, h), *Cdh12* KD (31%) did not impact CCK+ inputs targeting L5 ET neurons and *Cdh13* KD (36%) did not modify PV+ inhibition in L5 IT pyramidal cells (Fig. 3b, c, f, h). Altogether, our results demonstrate that 1) the function of postsynaptic *Cdh12* and *Cdh13* is cell type-dependent, and 2) is required for cell type- and input-specific inhibitory wiring onto L5 IT and L5 ET populations.

## Mapping pyramidal cell populations inhibitory connectivity

Our previous experiments revealed that the same cadherin molecule expressed in different cell types does not instruct the same inhibitory motifs. Interneuron diversity is extensive and several subtypes of PV+ and CCK+ basket cells have been identified[40]. Inhibitory connections mediated by CDH12 and CDH13 may thus rely on different presynaptic partners depending on the type of presynaptic interneuron involved. Based on this idea, we hypothesized that L5 IT and L5 ET neurons exhibit distinct inhibitory circuits composed of different interneuron subtypes and organized around the expression of matching cell-surface molecules. To test this hypothesis, we engineered a multiplex rabies-based monosynaptic tracing approach[41,42] to delineate the cell-type-specific inhibitory input maps of each

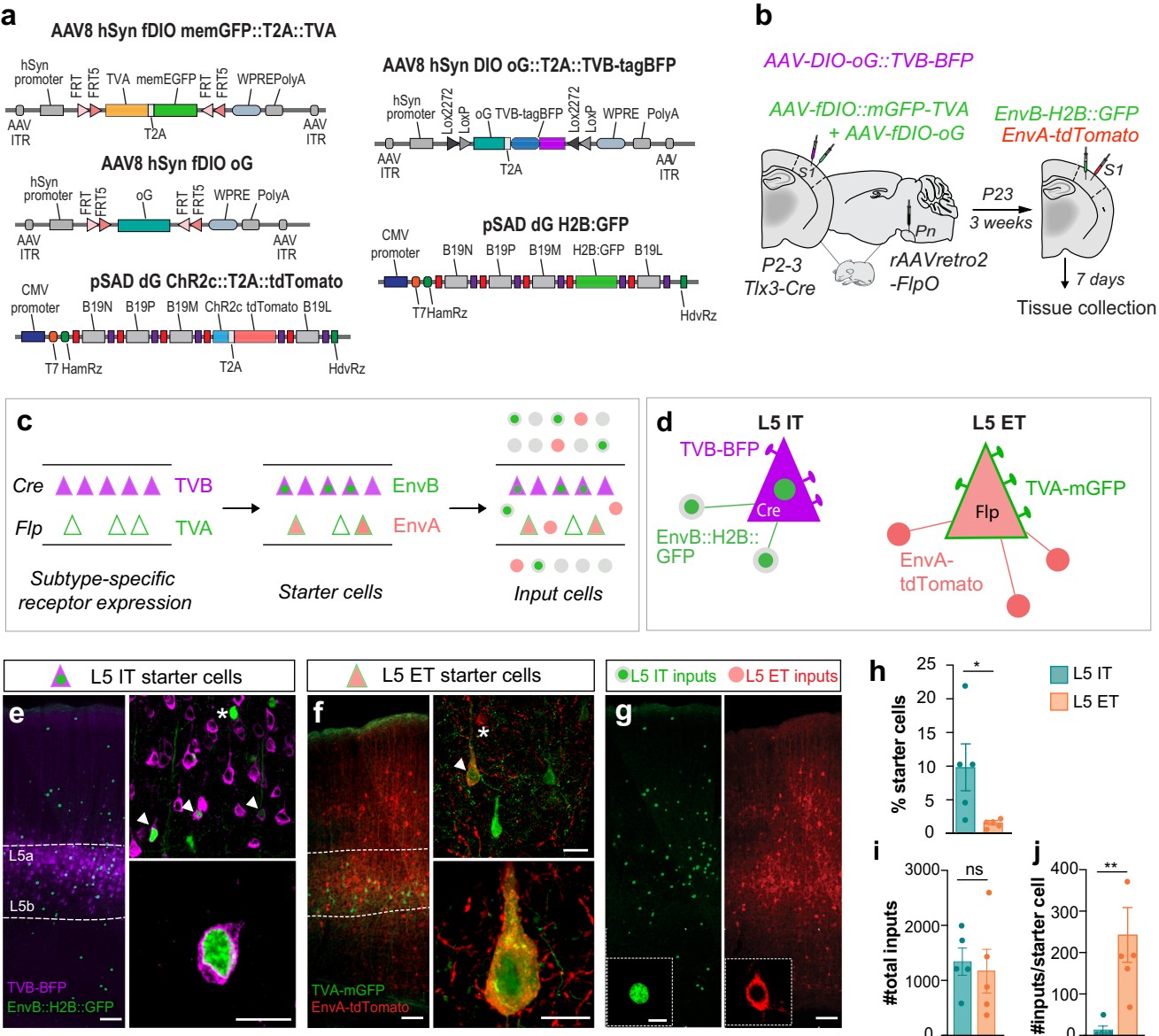

**Fig. 4 | Mapping L5 pyramidal cell populations inhibitory connectivity. a** Viral *Cre*- and *FlpO*-dependent oEnvA/oTVA and oEnvB/oTVB constructs. **b** Multiplex monosynaptic tracing strategy. **c** Schematic of the spatial distribution of receptor-expressing cells, starter cells and input cells for L5 IT and L5 ET populations. Note that L5 IT cells present in L5a express the *Cre* recombinase and can be infected with the TVB-receptor (magenta triangle) and/or EnvB (green and magenta triangle). L5 ET cells present in L5b express the *FlpO* recombinase and can be infected with the TVA-receptor (green outlined triangle) and/or EnvA (green and red triangle). Inputs from L5 IT (green circle) and L5 ET (red circle) neurons can then be mapped across the different cortical layers. **d** Schematic representation of L5 IT and L5 ET starter cells and their direct inputs expressing the different viruses. **e** Representative images of the spatial distribution of L5 IT input and starter cells. Top panel: L5 IT starter cells (white arrowhead) co-express TVB-BFP (magenta) and EnvB-H2B:GFP

(green). L5 IT input cells only express EnvB-H2B:GFP (asterisk). Bottom panel: L5 IT starter cell. **f** Representative images of the spatial distribution of L5 ET input and starter cells. Top panel: L5 ET starter cells (white arrowhead) co-express TVA-mGFP (green) and EnvA-tdTomato (red). L5 ET input cells only express EnvA-tdtTomato (asterisk). Bottom panel: L5 ET starter cell. **g** Representative images of L5 IT (green) and L5 ET (red) input cells distribution across a cortical column. **h** Fraction of L5 IT and L5 ET starter cells (*L5 IT, L5 ET n* = 5 mice, ratio paired t-test *$p$ = 0.0375). **i** Number of L5 IT and L5 ET total inputs (*L5 IT, L5 ET n* = 5 mice, ratio paired t-test $p$ = 0.5091). **j** Number of L5 IT and L5 ET inputs per starter cell (*L5 IT, L5 ET n* = 5 mice, ratio paired t-test **$p$ = 0.0045). Data are mean ± SEM, ns not significant. Scale bars: 50 μm (**e**–**g**), 25 μm (**e**–**g**), 5 μm (**g**). Source data are provided as a Source Data file.

population of L5 pyramidal cells (Fig. 4a). Multiplex monosynaptic tracing relies on specific rabies viral envelopes (EnvX) and their corresponding receptors (TVX). To simultaneously visualize presynaptic networks of L5 IT and L5 ET neurons, we designed distinct *EnvA/TVA* and *EnvB/TVB* complexes and employed two recombinase systems to selectively target L5 IT and L5 ET cells in the same animal (Fig. 4a–d). We injected a *FlpO*-expressing retrograde AAV in the pons of *Tlx3-Cre* mice; hence neighboring L5 ET and L5 IT neurons would express *FlpO* and *Cre*, respectively (Fig. 4a–d). We injected Flp-

and Cre-dependent viruses in S1 to obtain cell-type-specific receptor expression. To target L5 ET neurons, we injected two *FlpO*-dependent AAVs, one encoding a mutant TVA[66T] and a membrane-bound GFP and another encoding the RV glycoprotein *oG* (Fig. 4a–d). We then injected *Cre*-dependent AAVs expressing a tagBFP-TVB and *oG* to target L5 IT cells (Fig. 4a–d and Supplementary Fig. 9a, b). Three weeks post-injections, we injected two distinct rabies into S1; an EnvA-coated G-deficient rabies virus (RVΔG) expressing the tdTomato reporter and an EnvB-coated RVΔG expressing the histone H2B-

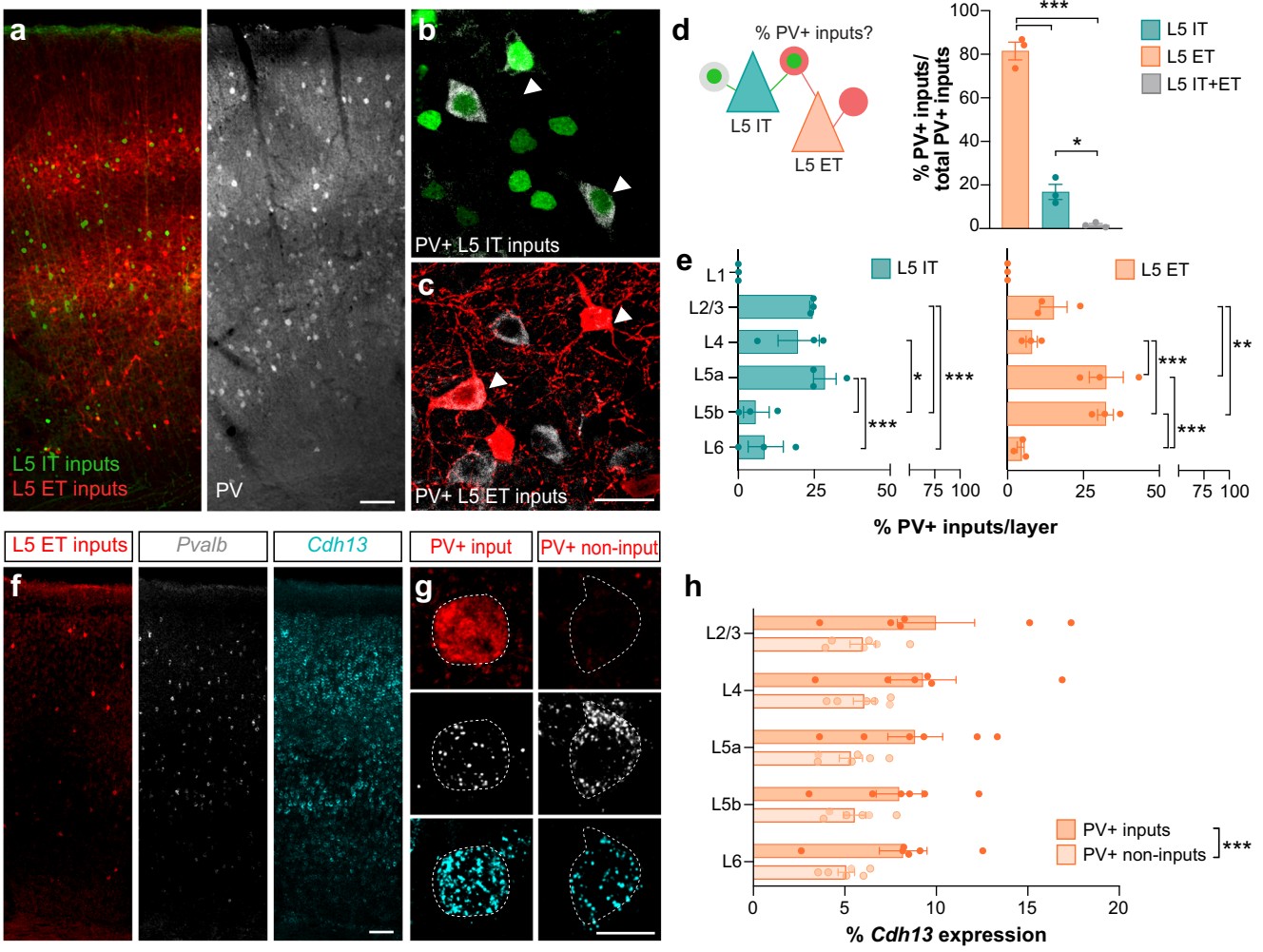

**Fig. 5 | L5 IT and L5 ET neurons are targeted by distinct populations of PV+ interneurons. a** Distribution of L5 IT (green) and L5 ET (red) input cells as well PV+ cells (gray) across cortical layers. L5 IT (**b**) and L5 ET PV+ input cells (**c**) are indicated with a white arrowhead. **d** Schematic of L5 IT and L5 ET connectivity. Proportion of PV+ inputs between L5 IT, L5 ET or L5 IT/ET PV+ inputs (*n* = 3 mice; One-way ANOVA \*\*\**p* < 0.001, Tukey's posthoc: *L5 IT* vs *L5 ET* \*\*\**p* < 0.001, *L5 IT + ET* vs *L5 IT* \**p* = 0.0317, *L5 IT + ET* vs *L5 ET* \*\*\**p* < 0.001). **e** Proportion of L5 IT and L5 ET PV+ inputs per cortical layer (*n* = 3 mice; Two-way ANOVA, Layer factor \*\*\**p* < 0.001, L5 population factor ns, Layer x L5 population interaction \*\**p* < 0.01. Results of the

Benjamini, Krieger and Yekutieli posthoc analysis are plotted on the graphs). **f** Distribution of L5 ET input cells alongside PV (*Pvalb*) and *Cdh13* RNA expression across cortical layers. **g** Representative images of *Pvalb* and *Cdh13* RNA expression in PV + L5 ET input and PV+ non-input cells. **h** *Cdh13* RNA expression across cortical layers in PV + L5 ET input and non-input cells (*n* = 6 mice; Two-way ANOVA repetitive measurements, Layer factor *p* = 0.8626, Input factor \*\*\**p* < 0.0001, Mouse factor \**p* = 0.0147, Layer x Input interaction *p* = 0.9422). Data are mean ± SEM, ns, not significant. Scale bars: 100 μm (**f**), 50 μm (**a**), 20 μm (**b, c**), 10 μm (**g**). Source data are provided as a Source Data file.

tagged nuclear GFP reporter (Fig. 4a–g). This combination allowed us to simultaneously trace and visualize inputs from both populations of L5 pyramidal cells (Supplementary Fig. 9a, b), which we examined 7 days after rabies injection.

As expected, L5 IT starter cells (expressing BFP and nuclear GFP) were mainly located in L5a and L5 ET starter cells (expressing membrane-bound GFP and cytosolic tdTomato) were restricted to L5b (Fig. 4e, f). We consistently observed more L5 IT starter cells than L5 ET starter cells (Fig. 4h), although the total number of presynaptic cells (or input cells) connecting to each L5 pyramidal cell population was comparable (Fig. 4i), validating our experimental approach. Furthermore, we did not observe any labeled cells in wild-type mice, further confirming the specificity of our method (Supplementary Fig. 9c). We then mapped the afferent connections of L5 IT and L5 ET starter cells throughout S1. In each brain, we quantified the number of input cells connected to each L5 pyramidal cell population as a fraction of the total number of input cells measured in S1 (Fig. 4i). We observed that the number of input cells per starter cell was much higher in the L5 ET population compared to the L5 IT (Fig. 4j), consistent with other

studies showing that L5 ET neurons receive more neuronal connections than L5 IT neurons[43].

We then assessed the identity of the presynaptic cells connected to L5 IT and L5 ET populations. We focused on PV+ interneurons 1) because they represent the most abundant source of perisomatic inhibition in the cerebral cortex[44–47] and 2) due to the incompatibility of our multiplex monosynaptic tracing system with the current tools allowing a reliable identification of cortical CCK+ basket cells. We thus quantified the proportion of PV+ input cells over the total number of input cells in S1 and found that L5 ET neurons receive more PV+ connections than L5 IT neurons (Fig. 5a–d). These results were consistent with our synaptic analysis, where L5 ET neurons receive more PV+ inputs (Fig. 1). Our multiplex monosynaptic tracing also revealed that the PV+ interneurons targeting L5 IT and L5 ET neurons exhibit distinct laminar distributions (Fig. 5a, e). We indeed observed that PV+ interneurons projecting to L5 IT neurons were spread across L2/3, L4 and L5a, while L5 ET neurons primarily received local inhibition from L5a and L5b PV+ cells (Fig. 5e). Surprisingly, we found a minimal fraction of double-labeled afferent PV+ cells primarily allocated in L5a

(Supplementary Fig. 9d–f). In sum, our experiments suggest that L5 IT and L5 ET neurons likely receive connections from different presynaptic networks of PV+ interneurons.

Finally, we explored whether different PV+ interneurons could target specific populations of L5 pyramidal neurons by expressing matching molecular programs. Based on this assumption, *Cdh13* that has been shown to establish homophilic interactions[48,49] should be enriched in PV+ interneurons targeting L5 ET neurons. Our analysis indeed revealed that *Cdh13* expression is significantly higher in PV+ cells connected to L5 ET neurons compared to other PV+ interneurons (Fig. 5f–h).

In summary, our findings reveal connectivity rules between pyramidal cells and PV+ interneurons and highlight the role of *Cdh12* and *Cdh13* as mediators of cell-type-specific perisomatic inhibition.

## Discussion

Understanding the core principles governing neuronal connectivity is essential to identifying the intricate flow of information across neuronal circuits in the cerebral cortex. Cortical interneurons exhibit a remarkable diversity and are critical for sculpting the information conveyed by pyramidal cells. Over the last years, new evidence has transformed our view on inhibition, shifting from unselective connectivity patterns[4–6,50] to precise inhibitory microcircuits whose rules are just beginning to be understood[7–13,15]. However, how this inhibition unfolds and the molecular programs underlying the emergence of precise cell-cell recognition are still unknown. Here, we explored the molecular programs orchestrating cell-type-specific perisomatic inhibition in the developing mouse somatosensory cortex. Our study demonstrates that pyramidal cells, through the expression of a unique cell-surface molecule signature, could shape basket cell interneuron wiring to drive cell-type-specific inhibitory connectivity. We found that distinct populations of L5 pyramidal cells promote unique perisomatic inhibitory patterns by expressing two cadherins. *Cdh12* conducts CCK+ inhibition onto L5 IT neurons, and *Cdh13* mediates PV+ inhibition onto L5 ET neurons. This differential inhibition signature is acquired during the initial steps of synaptogenesis and irrespective of the pyramidal cell's position, suggesting that connectivity features might be part of L5 cell type identity.

Several cadherin family members have been described as key synaptic regulators for cellular targeting in the retina and the hippocampus[32,33]. Here, we examined the postsynaptic expression of CDH12 and CDH13, showing that CDH13 is preferentially localized in the vicinity of PV+ synapses. Although we cannot discard the possibility that CDH12 and CDH13 ectopic expression does not recapitulate endogenous protein distribution, we did not observe any aberrant protein localization. Furthermore, both CDH12 and CDH13 showed a clustered membrane expression, suggesting their delivery and localization in the expected subcellular compartment. Our results are consistent with previous studies reporting pre and postsynaptic expression of CDH13 in inhibitory synapses[51–53]. The fact that CDH12 showed a homogeneous expression at the surface of L5 pyramidal somas might indicate additional roles of this protein.

Even though *Cdh12* and *Cdh13* are both expressed in L5 IT and L5 ET neurons, they exhibit different functions depending on the cellular context. Our genetic manipulations indeed demonstrate that CDH13 exclusively drives PV+ interneuron wiring onto L5 ET cells but does not play a similar role in L5 IT neurons. Conversely, CDH12 only instructs CCK+ basket connectivity in L5 IT cells despite also being expressed in L5 ET neurons. Why CDH12 and CDH13 roles are not conserved across different L5 pyramidal cell populations? It is conceivable that L5 IT and L5 ET neurons express different interactomes and may not contain the relevant postsynaptic machinery to allow CDH13 and CDH12 to instruct PV+ and CCK+ basket cell wiring, respectively. CDH13 is an atypical cadherin superfamily member as it lacks a transmembrane domain and is inserted into the plasma membrane via a glycosyl-phosphatidylinositol anchor[54]. As such, CDH13 requires other proteins for downstream signaling[55] and may interact with different postsynaptic partners depending on the pyramidal cell type. Identifying the co-receptors and downstream signaling mechanisms that regulate cadherin function in this cell-type-specific context would shed light on this process. Previous studies suggested that the chemokine CXCL12 might direct PV+ axons to target L5 ET neurons[56]. *Cxcl12* was not detected in our RNAseq screening (possibly due to its low expression), but we identified *Cdh12* and *Cdh13* as critical instructors of PV+ and CCK+ perisomatic inhibition onto L5 pyramidal cell populations such that *Cdh12* conducts CCK+ inhibition onto L5 IT neurons and *Cdh13* mediates PV+ inhibition onto L5 ET cells.

Both CDH12 and CDH13 are capable of homophilic interactions[48,49], and both pre and postsynaptic expression of CDH13 has been reported in inhibitory synapses[52,53]. CDH12 and CDH13 may also interact with different molecular partners depending on the presynaptic interneuron engaged. Trans-synaptic rabies tracing using glycoprotein (G)-deleted rabies virus is a powerful and versatile tool to explore circuit connectivity and uncover the distribution of inputs onto specific neuronal populations[57]. We engineered a multiplex monosynaptic tracing approach to visualize inhibitory inputs onto L5 IT and L5 ET cell types in the same brain, based on previous studies[41,42]. We found that L5 ET neurons receive a larger fraction of PV+ inputs per starter cell than L5 IT neurons, but also that the inhibitory presynaptic networks of L5 IT and L5 ET neurons are largely segregated. To our knowledge, such exclusive connectivity patterns have not been described before in the cerebral cortex and previous studies tend to indicate that neighboring cells share inputs[4–6,50]. We cannot rule out the potential competition between the two RVΔG or the incomplete mapping of all inhibitory inputs targeting L5 pyramidal cells may jeopardize our capacity to visualize shared PV+ inputs between L5 IT and L5 ET neurons. However, we could identify an obvious fraction of excitatory neurons infected with both RVΔG thus confirming that co-infections are biologically possible in our system.

Previous studies have highlighted the existence of translaminar PV+ inhibition in L5 pyramidal cells[58–61]. Our results revealed that L5 ET neurons receive inhibition primarily from local L5 PV+ interneurons, while L5 IT neurons receive inputs from PV+ cells spread across different layers, with similar local and translaminar inhibitory contributions. At least ten PV+ interneuron subtypes with unique transcriptomic profiles and laminar positions have been identified[40]. Hence, the laminar distribution of PV+ interneurons targeting L5 IT and L5 ET populations seem to reflect the existence of different inhibitory circuits organized by the expression of matching cell-surface molecules, and our data suggest that CDH13-CDH13 interactions could contribute to shaping cell-type-specific PV+ connectivity onto L5 ET neurons. Furthermore, the diverse expression pattern of *Cdh13* among PV+ interneurons, as well as the different spatial distribution of PV+ interneurons connected to L5 IT or L5 ET strongly suggest that segregated PV-to-pyramidal cell type connections are a rule of inhibitory connectivity in the cerebral cortex, at least for L5.

Extensive work has demonstrated the dichotomy of PV+ and CCK+ perisomatic inhibition[23,62]. Both basket cells exhibit different intrinsic properties and fire in different phases of behavioral activity[63], but some studies suggest that the function of these interneurons is interlinked. Experience-dependent or experimentally induced changes in the activity of pyramidal cells cause bidirectional changes in perisomatic inhibition in the hippocampus[64,65]. Specifically, increased activity in pyramidal cells enhances PV-mediated inhibition and reduces CCK-mediated inhibition[64,65]. Here, we found no compensatory increase between both types of perisomatic inhibition. Modifying the density of one type of presynaptic input – by interfering with *Cdh12* or *Cdh13* levels – did not impact the other, supporting the idea that PV+ and CCK+ basket cells have non-overlapping functions.

But why would some neurons need both types of perisomatic inhibition while others receive barely any CCK+ inhibition? L5 IT and L5 ET pyramidal cell populations consistently exhibit distinct functions across the different cortical areas[1,66–68]. Evidence supports the notion that L5 IT neurons uphold and enhance the computational efficiency of all cortical neurons by continuously updating them[69]. As such, they compute more diverse information and require increased plasticity[70]. CCK+ innervation may thus confer more precision in the modulation of L5 IT neuron activity, a phenomenon further reinforced by modulatory pathways[71,72]. In contrast, L5 ET neurons serve as information broadcasters, and thus need tighter inhibitory regulation in order to faithfully relay updated information to subcortical structures[69]. Indeed, mice demonstrate a more adept ability to regulate the activity of L5 ET neurons compared to L5 IT neurons[70]. Future directions should go on understanding the relevance of the different ratios of PV+ and CCK+ inhibition onto L5 IT and L5 ET pyramidal cells.

Considering the unique function of L5 pyramidal cells, it is foreseeable that any disruption or mistargeting of their inhibitory inputs could significantly affect the circuitry and lead to various pathological conditions[73]. For instance, L5 pyramidal cells seem particularly vulnerable in autism[74,75]. Future investigations to understand the relationship between synaptic excitatory/inhibitory ratio within cortical circuits, input specificity and risk gene expression might shed some light on the etiology of some neurodevelopmental disorders.

## Methods

### Animals

*Tlx3*[Cre38] and *Nex*[Cre76] mice were maintained in a C57BL/6 background (Charles River Laboratories). Animals were housed in groups of up to five littermates and maintained under standard, temperature controlled, laboratory conditions. Mice were kept on a 12:12 light/dark cycle and received water and food *ad libitum*. For all experiments, both females and males were used and were randomly allocated to experimental groups. All animal procedures were approved by the ethical committee (King's College London) and conducted in accordance with European regulations and Home Office personal and project licenses (PPL70/8322, PD025E9BC, PP3627598) under the UK Animals (Scientific Procedures) 1986 Act.

### Plasmids design

**shRNA**. *Cdh12* and *Cdh13 shRNA* were cloned into a *pAAV-EF1a-DIO-mCherry* vector as previously described[37]. The ssDNA primers to generate the *shRNAs* were obtained using the Block-it RNAi web tool (Thermo Scientific) and were as follows: *shCdh12#1* (Fwd: CTA GGC ATT CGG ACT TGG ATA AAG GCC TGA CCC ACC TTT ATC CAA GTC CGA TGC TTT TTG and Rev: AAT TCA AAA AGC ATT CGG ACT TGG ATA AAG GTG GGT CAG GCC TTT ATC CAA GTC CGA ATGC); *shCdh12#2* (Fwd: CTA GGC AGT ACC AGG TCC TCA TTC ACC TGA CCC ATG AAT GAG GAC CTG GTA CTG CTT TTTG and Rev: AAT TCA AAA AGC AGT ACC AGG TCC TCA TTC ATG GGT CAG GTG AAT GAG GAC CTG GTA CTGC; *shCdh12#3* (Fwd: CTA GGC TGG GCC ATT AAA GGA TAC TCC TGA CCC AAG TAT CCT TAA ATG GCC CAG CTT TTTG and Rev AAT TCA AAA AGC TGG GCC ATT AAA GGA TAC TTG GGT CAG GAG TAT CCT TAA ATG GCC CAGC); *shCdh12#4* (Fwd: CTA GGC AAT TCT CCT TTA GAT TAG CCC TGA CCC AGC TAA TCT AAA GGA GAA TTG CTT TTTG and Rev: AAT TCA AAA AGC AAT TCT CCT TTA GAT TAG CTG GGT CAG GGC TAA TCT AAA GGA GAA TTGC); *shCdh12#5* (Fwd: CTA GGC ACG AAT ACA ATG ACTAT TCC CTG ACC CAG AAT AGT CAT TGT ATT CGT GCT TTT TG and Rev: AAT TCA AAA AGC ACG AAT ACA ATG ACT ATT CTG GGT CAG GGA ATA GTC ATT GTA TTC GTGC); *shCdh13#1* (Fwd: CTA GGC TCC TTG CAG GAT ATC TTT ACC TGA CCC ATA AAG ATA TCC TGC AAG GAG CTT TTTG and AAT TCA AAA AGC TCC TTG CAG GAT ATC TTT ATG GGT CAG GTA AAG ATA TCC TGC AAG GAGC); *shCdh13#2* (Fwd: CTA GGG GCT GCA TAC ACC ATC ATC ACC TGA CCC ATG ATG ATG GTG TAT GCA GCC CTT TTTG

and Rev: AAT TCA AAA AGG GCT GCA TAC ACC ATC ATC ATG GGT CAG GTG ATG ATG GTG TAT GCA GCCC); *shCdh13#3* (Fwd: CTA GGC TGA TCA AAG TGG AGA ATG ACC TGA CCC ATC ATT CTC CAC TTT GAT CAG CTT TTTG and Rev: AAT TCA AAA AGC TGA TCA AAG TGG AGA ATG ATG GGT CAG GTC ATT CTC CAC TTT GAT CAGC); *shCdh13#4* (Fwd: CTA GGC CTT CTT CAG AAT CTG AAC ACC TGA CCC ATG TTC AGA TTC TGA AGA AGG CTT TTTG and Rev: AAT TCA AAA AGC CTT CTT CAG AAT CTG AAC ATG GGT CAG GTG TTC AGA TTC TGA AGA AGGC); *shCdh13#5* (Fwd: CTA GGC CCA TCA TGG TGA CAG ATT CCC TGA CCC AGA ATC TGT CAC CAT GAT GGG CTT TTTG and Rev: AAT TCA AAA AGC CCA TCA TGG TGA CAG ATT CTG GGT CAG GGA ATC TGT CAC CAT GAT GGGC). *shCdh12#2* and *shCdh12#4* were selected for the *Cdh12* KD experiment. *shCdh13 #2* and *shCdh13#4* for were selected for the *Cdh13* KD experiment. The same *shRNA* against *LacZ (shLacZ)* as described in ref. 37 was used as a control condition. We did not perform any rescue experiment due to the large size of *Cdh12* and *Cdh13* sequences, which made it challenging to pack them into a single AAV with shRNA sequences.

**Rabies**. Constructs used for the multiplex G-deleted rabies virus-mediated circuit mapping approach were generated by standard molecular biology cloning procedures. The *pSAD ΔG H2B:GFP* was obtained by taking advantage of the *pSAD ΔG F3* expression vector (kindly provided by M. Tripodi, Cambridge University, AddGene #32634). The nucleotide sequence encoding the histone H2B-tagged GFP was designed and ordered via GeneArts (ThermoFisher Scientific). Subsequently, it was subcloned within the *pSAD ΔG F3* expression vector by a restriction enzyme-based method, using the unique restriction enzymes AscII and BsiWI (New England Bioscience). The membrane-bound *EGFP::T2A::TVA* cassette was obtained by gene synthesis. A DNA fragment containing the nucleotide sequence of the membrane-bound EGFP link via a T2A cleaving site to the TVATC66 receptor Field 69 coding sequence was designed in an inverted orientation and ordered via GeneArts (ThermoFisher Scientific). This DNA sequence was subsequently amplified by Q5 Hot Start High Fidelity DNA polymerase-based PCR (New England Bioscience) using primers containing the unique restriction sites AccI and NheI (Fwd: TAA GCA GTC GAC TTA CTT GGA TGC GCT TTC AAG and Rev: TGC TTA GCT AGC GCC ACC ATG CTG TGC TGT ATG). The amplified sequence was then inserted within the expression vector *pAAV-hSyn-fDIO-MCS* (kindly provided by M. Selten and O. Marín, King's College London) using the AccI and NheI sites.

The same experimental approach has been used to generate the *pAAV-hSyn fDIO oG* expression vector. The oG nucleotide sequence was PCR amplified using specific primers containing the restriction sites AccI and NheI (Fwd: TAA GCA GTC GAC TTA GAG CCG TGT CTC GCC and Rev: TGC TTA GCT AGC GCC ACC ATG TCC CCA GCT CTC CTC) and then cloned into *pAAV-hSyn-fDIO-MCS*. The *pAAV-TVB-tagBFP::T2A::oG* was designed as previously described[42] and ordered via GeneArts (ThermoFisher Scientific). The cassette was then subcloned into the expression vector *pAAV-hSyn Flex tdTomato::T2A::SypEGFP* (AddGene #51509) using the AscI and FseI restriction sites. All constructs were verified by sequencing.

### Cell culture and transfection

HEK293T cells were cultured in Dulbecco's Modified Eagle's medium supplemented with 10% fetal bovine serum, 2 mM glutamine, penicillin (50 units/ml) and streptomycin (50 g/ml). The cultures were incubated at 37 °C in a humidified atmosphere containing 5% $CO_2$. HEK293T cells were transfected using polyethylenimine (PEI, Sigma) at a 1:4 DNA:PEI ratio or Lipofectamine 2000 (Thermo Fisher Scientific).

### AAV and rAAVretro production

HEK293FT cells (Thermo Fisher Scientific R70007) were seeded on 15-cm plates and co-transfected with packaging plasmids AAV-ITR-2

genomic vectors (7.5 μg), AAV-Cap8 vector pDP8 (30 μg; PlasmidFactory GmbH, Germany, #pF478) or AAV-Cap DJ Rep-Cap vector (30 μg; Cell Biolabs, VPK-420-DJ) using PEI (Sigma) at a ratio 1:4 (DNA:PEI). 72 h post-transfection, supernatants were incubated with Ammonium sulfate (65 g/200 ml supernatant) for 30 min on ice and centrifuged for 45 min at 4000 RPM at 4 °C. Transfected cells were harvested and lysed (150 mM NaCl, 50 mM Tris pH8.5), followed by three freeze-thaw cycles and Benzonase treatment (50U/ml; Sigma E1014-25KU) for 1 h at 37 °C. Filtered AAVs (0.8 μm and 0.45 μm MCE filters) from supernatants and lysates were run on an Iodixanol gradient by ultracentrifugation (Vti50 rotor, Beckmann Coultier) at 50,000 RPM for 1 h at 12 °C. The 40% iodixanol fraction containing the AAVs was collected, concentrated using 100 kDa-MWCO Centricon plus-20 and Centricon plus-2 (Merck-Millipore), aliquoted and stored at −80 °C. The number of genomic copies was determined by qPCR using the following primers against the WPRE (Fwd: GGC ACT GAC AAT TCC GTG GT and Rev: CGC TGG ATT GAG GGC CGAA). AAVs with a titer equal or higher to $10^{12}$ genome copy/ml were used for in vivo injections. For down-regulation experiments, the two shRNAs with the most efficient down-regulation in vitro were used for each target gene. The two shRNA plasmids were co-transfected for AAV production to increase the down-regulation efficiency.

## Rabies production
**Cell lines.** HEK-TVA (The Salk Institute of Biological Sciences), HEK-TVB (The Salk Institute of Biological Sciences), BHK-EnvA (kindly gift from T. Karayannis, University of Zurich), BHK-EnvB (The Salk Institute of Biological Sciences), and B7GG (kind gift from T. Karayannis) cells were maintained in DMEM (Gibco), supplemented with 10% fetal bovine serum (FBS) in a humified atmosphere of 3% $CO_2$ and 35 °C.

**G-deleted rabies virus production.** G-deleted rabies viruses (RVΔG) were produced as previously described[41]. Briefly, *RVΔG-H2B:GFP* and *RVΔG-tdTomato* were recovered in B7GG cells by Lipofectamine 2000 (Thermo Fisher Scientific) transfection with *pcDNA-SADB19N*, *pcDNA-SADB19P*, *pcDNA-SADB19L* and *pSADΔG H2B:GF*P or *pSADΔG:tdTomato* (kind gift from K. Conzelmann). During the virus production, the transfected cells were maintained using DMEM medium (Gibco) supplemented with 10% FBS in a humified atmosphere of 3% $CO_2$ and 35 °C.

For pseudotyping with EnvA and EnvB, BHK-EnvA and BHK-EnvB cell lines were infected with unpseudotyped *SAD ΔGtdTomato* and *SAD ΔGH2B:GFP* rabies viruses, respectively. Subsequently, infected cells were washed with PBS, collected by 0.25% trypsin-EDTA and replated in new dishes. The virus-containing medium was then filtrated via 0.45 μm filters (Cornings) and concentrated through two rounds of ultra-centrifugation. The infectious titers of the purified viruses were determined using the HEK-TVA and HEK-TVB cell lines. In addition, the presence of any contamination of unpseudotyped rabies virus was detected by using HEK293T cells. Aliquots containing pseudotyped rabies viruses were stored at −80 °C.

## Stereotaxic injections
**Retrobeads.** L5 pyramidal cell populations were targeted using green (488 nm) or red (555 nm) fluorescent retrobeads IX (Lumafluor Corp., FL). P2-3 pups were anesthetized with isoflurane (2.5%) and mounted on a stereotactic frame using a 3D printed isoflurane mask. Unilateral injections of 75 nl retrobeads at 30 nl/min were carried out as follows: L5 IT were labeled by targeting the contralateral somatosensory cortex (S1) (AP +1.6, ML −1.9 to −2.2, DV −0.8 to −0.5), L5 ET were labeled by targeting the Pons (AP −0.3, ML + 0.3, DV −4.5 to −4.0). Retrobeads were sonicated prior to each injection to avoid aggregate formation. The pipette was retracted from the brain after 2 min to allow for diffusion.

**AAV viral injections.** For in situ hybridization experiments, we followed the same experimental design as previously described to target L5 IT and L5 ET cell types. Briefly, 300 nl of *rAAV2retro-Ef1a-tagBFP* ($1,02.10^{12}$ vg/ml) and 300 nl of *rAAV2retro-Ef1a-NLS-tdTomato* ($4,82.10^{12}$ vg/ml) were injected at 60 nl/min in the contralateral S1 and ipsilateral Pons of P2/3 WT pups.

For *Cdh12* and *Cdh13* down-regulation experiments, we injected 300 nl of *AAV8-Ef1a-DIO-shCdh12-mCherry* ($8,20.1^{11}$ vg/ml) or *AAV8-Ef1a-DIO-shCdh13-mCherry* ($9,1.10^{11}$ vg/ml) or *AAV8-DIO-shLacZ-mCherry* ($1,20.10^{12}$ vg/ml) in the S1 of P2/3 pups. *Tlx3-Cre* mice were used to specifically access L5 IT, while L5 ET were specifically targeted by injecting 300 nl of *rAAV2retro-Cre* (AddGene #55636-AAVrg, $2.10^{13}$ vg/ml) in the Pons. The stereotaxic coordinates for the different target areas were the same as for the retrobeads experiment.

**Rabies stereotaxic injections.** P2-3 *Tlx3-Cre* pups were anesthetized with isoflurane (2.5%) and mounted on a stereotactic frame using a 3D-printed isoflurane mask. A unilateral injection of 300 nl *rAAV2retro-FlpO* (AddGene #55637-AAVrg, $1,6.10^{13}$ vg/ml) was made in the Pons (AP −0.3, ML + 0.3, DV −4.4 to −4.0) to target L5 ET axons. 200 nl of a mix of *AAV8-DIO-oG* ($1,65.10^{12}$ vg/ml) and *AAV8-DIO-hSyn-TVB-BFP* ($4,37.10^{13}$ vg/ml) (2:1 mix ratio), and 200 nl of a mix of AAV8-FRT-oG ($9,61.10^{13}$ vg/ml) and *AAV8-FRT-TVA-mGFP* ($9,87.10^{13}$ vg/ml, 2:1 mix ratio) were co-injected in the ipsilateral somatosensory cortex (S1, AP + 1.6, ML −1.9 to −2.2, DV −0.8 to −0.5). All viral injections were made at a 60 nl/min rate using a Micro2T nanoinjector (WPI). The pipette was retracted from the brain after 5 min, and the wound was closed using VetBond glue. Three weeks later, a second injection of 500 nl of a mix of *RVΔG-EnvB-H2B:GFP* ($1,6.10^{10}$ TU/ml) and *RVΔG-EnvA-tdTomato* ($3,65.10^8$ TU/ml, kind gift from A. Delogu) (1:1 ratio) was made in the ipsilateral S1BF (AP-1.8, ML 3.2, DV −0.7 --> −0.4). After suturing and disinfecting with Betadine, mice received a subcutaneous injection of Buprenorphine (0.03 mg/ml) to prevent acute pain. Note that the stereotaxic coordinates for the different target areas were determined from the lambda in pups (Atlas of the Developing Mouse Brain[77]) and the bregma in young adults[78].

## Tissue dissociation and Fluorescence-Activated Cell Sorting (FACS)
P2-3 C57BL/6 mice were injected, as previously described, with Red Retrobeads™ IX (RB555, LumaFluor) either in S1BF to target L5 IT, or in the pons to label L5 ET neurons. We used the same fluorophore for both L5 populations to ensure comparable detection parameters. L5 IT and L5 ET populations were thus sorted in independent experiments. To isolate individual cells, P10 mice were killed by decapitation, the brain was extracted and lower layers of S1 were microdissected in cold pH 7.3 dissociation media containing 14 mM $MgCl_2$, 2 mM HEPES (Invitrogen 15630-106), 0.2 mM NaOH (Sigma S0899), 90 mM $Na_2SO_4$ (Sigma S6547), 30 mM $K_2SO_4$ (Sigma P9458), 3.6 mg/mL D-(+)-Glucose (Sigma G6152), 0.8 mM kynurenic acid (Sigma K3375), 50 μM AP-V (Sigma A5282), 50 U/mL penicillin/streptomycin (Thermo Fisher 15140122). To generate single-cell suspensions, 1 mm³ tissue pieces from 2 to 3 brains were pooled and enzymatically digested in a dissociation medium containing 0.16 mg/mL cysteine (Sigma C9768), 7 U/mL Papain (Sigma P3125), 0.1 mg/mL DNase (Sigma 10104159001) at 37 °C for 30 min. Papain digestion was then blocked with a dissociation medium containing 0.1 mg/mL ovomucoid (Sigma, St. Louis, MO T2011) and 0.1 mg/mL bovine serum albumin (BSA, Sigma A4161) for 1 min at room temperature. Neurons were mechanically dissociated to create a single cell suspension in iced OptiMEM solution containing 3.6 mg/mL D-(+)-Glucose, 4 mM $MgCl_2$, 0.4 mM kynurenic acid, 25 μM AP-V, 0.04 mg/mL DNase, diluted in OptiMEM medium (Thermo Fisher 31985). Actinomycin D (Sigma A1410) was added during the dissociation process to protect the tissue and prevent activation of immediate early genes[79]. Cells were centrifugated at $120 \times g$ for 5 min at 4 °C,

resuspended in 150–300 μL of fresh complemented OptiMEM and passed through a 40 μm cell strainer. Retrobeads-labeled (RB555+) individual cells were then purified from the cell suspension on a BD FACS Aria III flow cytometer using FACSDiva software with optimal compensation and gain settings determined for each experiment. Most FACs were performed in the afternoon. No significant difference was found across the collection times. Doublets were excluded based on SSC-A vs. SSC-W plots and FSC-A vs. FSC-W plots. Live cells were gated based on FSC-A and the exclusion of cells positive for DAPI. Finally, live RB555+ cells were selected based on FITC-A vs PE-A plots, and were collected in 350 μl of RLT buffer (RNeasy Lysis buffer, QIAGEN) containing 1% 2-Mercaptoethanol and stored at −80 °C for RNA extraction.

## RNA-sequencing
RNA was extracted using the QIAGEN RNeasy Micro Kit according to the manufacturer's instructions. Library preparation and RNA-seq experiments were performed by the Genomic Unit of the Centre for Genomic Regulation (CRG, Barcelona, Spain). Approximately 10,000–20,000 cells were required to obtain 5–10 ng of total RNA, which served as input for the library preparation using the SMARTer v4 Ultra Low RNA Kit. The Illumina HiSeq 2500 platform was used to sequence libraries to a mean of approximately 100 million mapped 125 base pair paired-end reads per sample. In the RNA-seq experiments, three biological replicates were ascertained for each dataset.

## RNA-sequencing deconvolution analysis
The proportions of L5 IT, L5 ET and microglia cell types were estimated using the R package MuSiC (https://www.nature.com/articles/s41467-018-08023-x). RNA-seq gene expression counts from this study were compared with mouse single-cell RNA-seq data from adult cortices (GSE185862). Count matrices were firstly converted to the ExpressionSet object format and followed by cellular deconvolution using MuSiC's music_prop function.

## RNA-sequencing data processing and differential expression analysis
Sequencing files from this study and ref. 27 were QC'ed, genome aligned, and gene expression quantified using Nextflow's RNA-seq pipeline (v3.2). Ensembl's GRCm38 annotation and genomic sequences were used as reference. Gene expression counts data were kindly provided by D. Jabaudon (Geneva University) from ref. 26, and an unpublished dataset of L5 ET cortical spinal cord pyramidal cells. Genes with at least 10 read counts in each sample replicated across all studies were kept. Differential gene expression analysis was performed to identify differentially expressed genes (DEGs) between L5 IT and L5 ET neurons in each study using the R/Bioconductor package DESeq2. Genes that showed a 1.5-fold change in expression and passed the 0.05 Bonferroni-adjusted p-values were labeled as differentially expressed. Our differential expression analyses revealed a contamination of microglial genes in our L5 ET samples. Although markers for L5 ET neurons were still enriched, this contamination is predicted to affect the read sampling of poorly expressing genes and its downstream differential analysis. To obtain a clearer L5 ET profile, we combined our dataset with two reference datasets that characterized L5-specific genes in early postnatal development[26,27]. We obtained the union of differentially-expressing genes from our study and[26], in which we retained genes that were already differentially expressed at an earlier time point[27] for downstream analyses.

## Gene Ontology analysis
Functional enrichment in gene functions was determined using the R/Bioconductor package ClusterProfiler. The org.Mm.eg.db package containing Gene Ontology terms was used as the database, and a list of stably expressed genes was used as background reference. Terms that passed the 0.05 FDR threshold were annotated as significantly enriched.

## Scoring of cell-adhesion molecules
A list of ligand-receptor pairs of cell-surface molecules was manually compiled from previous studies[28–30,34] and from the STRING database. Fold-change score of the expression of L5 IT- and L5 ET-specific adhesion molecules were calculated from the expression values from this and[26] studies. Essentially, the fold-difference in mean normalized gene expression between L5 IT and L5 ET cell types was calculated and scaled to produce values between 0 and 1. A specificity score assessing the specificity of the interaction between L5 IT with CCK+ interneurons, and between L5 ET with PV+ interneurons was determined by calculating the enrichment in the gene expression of partner adhesion molecules amongst interneuron subpopulations. Gene expression profiles of single cells from ref. 80 were used for this part of the analysis. CCK+ interneurons were identified in the Mouse Brain Atlas by a high expression of the *Cck* gene and moderate/low expression of the *Pvalb* gene. Essentially, log2 fold-difference in the expression of each receptor in CCK+ or PV+ interneurons was calculated by comparing its expression to other interneuron subpopulations. Receptor genes that are positively enriched in CCK+ or PV+ cells were retained and log2 fold-difference values from each ligand-receptor pair were averaged to give a representative specificity score.

## Single-molecule fluorescence in situ hybridization
For candidate gene validation in L5 pyramidal neurons and interneurons, P10 WT mice co-injected with *rAAV2-retro-tagBFP* and *rAAv2retro-tdTomato* were perfused as previously described. Brains were post-fixed overnight in 4% PFA in PBS, followed by cryoprotection in 30% sucrose-RNase-free PBS, and finally sectioned frozen on a sliding microtome at 30 μm. For down-regulation and rabies-related experiments, 40 μm slices from P30 mice were used. Fluorescent in situ hybridization on brain slices was performed according to manufacturer's protocol (ACD Bio, RNAscope Multiplex Fluorescent Assay v2, #323110). *Cdh12* (ACD Bio, #842531), *Cdh13* (ACD Bio, #443251-C3), *Pvalb* (ACD Bio, #421931), *tag-BFP* (ACD Bio, #537141-C2), *Cck* (ACD Bio, #402271-C3), *Sncg* (ACD Bio, #482741-C2) probes were used and visualized with Opal 520 (Akoya BioScience, FP1487001KT), Opal 570 (Akoya BioScience, FP1488001KT) and Opal 650 (Akoya BioScience, FP1496001KT). Single-molecule dual-color fluorescent in situ hybridization was combined with immunohistochemistry, and the following primary and secondary antibodies were used: chicken anti-GFP (1:100, Aves Lab GFP-1020), rabbit anti-tagRFP (1:100, Evrogen AB233), goat anti-chicken biotin (1:200, Vector BA-9010), Streptavidin 555 (1:400, Molecular Probes S32355), goat anti-chicken 488 (1:600, Molecular Probes A11039), goat anti-chicken 568 (1:500, ThermoFisher A11041) and donkey anti-rabbit 405 (1:200, Abcam Ab175652).

## In utero electroporation
Cortices of E12.5-day-old embryos (E12.5) of timed-pregnant *Nex^Cre* mice were unilaterally electroporated with EGFP::T2A::HA_CDH13 or EGFP::T2A::HA_CDH12. Briefly, timed-pregnant females were deeply anesthetised with isoflurane (2.5–5%) and a subcutaneous injection of buprenorphin (0.05 mg/kg) was provided after surgery. The uterus was exposed and 1 μl of a solution of 1.5μg/μl DNA and 0.01% Fast Green dye was injected in the lateral ventricle using a bevelled glass capillary. The embryo's head was carefully placed between the paddles of pair of platinum tweezer-type electrodes (Nepa Gene), while the third electrode was precisely located to target the somatosensory region of the cortex. Five pulses (amplitude: 25 V, duration: 50 ms, interval: 950 ms) were delivered using an electroporator (Nepa 21 Super Electroporator, Nepa Gene). The uterus was then moved back to its natural position, the incision sutured closed and the dam permitted to give birth normally.

## Immunohistochemistry

Animals were deeply anesthetized with an overdose of sodium pentobarbital (intraperitoneal injection) and transcardially perfused with 0.9% NaCl followed with 4% paraformaldehyde (PFA) in PBS. Dissected brains were post-fixed for 2 h at 4 °C, cryoprotected successively in 15% and 30% sucrose (Sigma S0389) in PBS, and finally cut frozen on a sliding microtome (Leica SM2010 R) at 40 μm. Free-floating brain slices were permeabilized with 0.25% Triton X-100 (Sigma T8787) in PBS for 1 h at room temperature (RT) and blocked for 2 h in a solution containing 0.3% Triton X-100, 1% serum, and 5% bovine serum albumin (BSA) (Sigma A8806) at RT. Brain slices were then incubated overnight at 4 °C with primary antibodies. The next day, the tissue was repeatedly rinsed in PBS and incubated with secondary antibodies for 2 h at RT. All primary and secondary antibodies were diluted in 0.3% Triton X-100, 1% serum and 2% BSA.

For perisomatic inputs quantification, the following primary and secondary antibodies were used: mouse anti-NeuN (1:500, Sigma MAB377), rabbit anti-NeuN (1:500, Millipore ABN78), goat anti-CB1R (1:400, Frontier Institute CB1-Go-Af450-1), mouse anti-CB1R (1:500, SySy 258011), rabbit anti-DsRed (1:500, Clontech 632496), mouse anti-Syt2 (1:125, ZFIN ZDB-ATB-081002-25), chicken anti-GFP (1:1000, Aves Lab GFP-1020), rabbit anti-GFP (1:500, Molecular Probes A11122), rat anti-HA (1:500, Sigma 27573500) donkey anti-rabbit 405 (1:250, Abcam Ab175652), donkey anti-goat 488 (1:500, Molecular Probes A11055), goat anti-mouse IgG2b 488 (1:500, Molecular Probes A21141), goat anti-mouse IgG2b 555 (1:500, Molecular Probes A21147), goat anti-mouse IgG2a 555 (1:500, Molecular Probes A21137) donkey anti-mouse 647 (1:500, Molecular Probes A31571), goat anti-rat Cy5 (1:500, Molecular probes A10525) and goat anti-mouse IgG2a 647 (1:500, Molecular Probes A21241).

For mapping of L5 IT and L5 ET monosynaptic inputs, every 3 sections containing somatosensory cortex were used (8 sections per mouse) and incubated with the following primary and secondary antibodies: guinea pig anti-RFP (1:500, SySy 390004 for tdTomato amplification), rabbit anti-tagRFP (1:250, Evrogen AB233 for BFP amplification), chicken anti-PV (1:250, SySy 195006), donkey anti-chicken 405 (1:200, Jackson 703-475-155), goat anti-guinea pig 555 (1:500, Molecular Probes A21435) and donkey anti-rabbit 647 (1:500, Molecular Probes A31573). The TVA-mGFP and EnvB-nGFP endogenous fluorescence was not amplified to avoid any cross-reaction with the BFP expression.

## Image acquisition and analysis

**Perisomatic input density analysis.** Confocal z-stacks (0.2 μm step size, 1024 × 1024, 8 bits) were acquired with an ×100 objective (1.44 NA) on an inverted Leica TCS-SP8 microscope and analyzed with IMARIS 7.5.2 software. All z-stacks were submitted to a background subtraction (13.2 μm) and a Gaussian filter (0.0517 μm) step prior analysis. L5 IT (imaged in L5a and L5b) and L5 ET (imaged L5b) somatas were 3D-reconstructed using the "Create surface" tool. SYT2+ and CB1R+ boutons were automatically detected using the "Spots" tool, and the number of SYT2+ and CB1R+ perisomatic boutons was then quantified using the "Find spots close to surface" tool (ImarisXT extension). Note that the volume of reconstruction could vary with the labeling method (viral cytosolic mCherry expression *versus* endogenous NeuN staining, but all experimental conditions are compared to their respective controls). XY filter (bouton size) and threshold distance (i.e. distance between the surface of the reconstructed soma and the center of the detected bouton) parameters were designed for each input type, according to their respective axonal bouton size and morphology. SYT2+ boutons were detected with a spot diameter of 0.8 μm and a distance threshold of 0.4 μm was applied to only select boutons in contact with the soma as described before[37]. Since CB1R+ boutons are larger than SYT2+ boutons, we used a spot diameter of 1.2 μm and a

distance threshold of 0.8 μm. Somas without SYT2+ and/or CB1R+ perisomatic inputs were excluded from the analysis.

**Cadherin localization.** Confocal z-stacks (0.1 μm step size) of electroporated L5 pyramidal neurons were taken on an inverted Leica TCS-SP8 confocal with a ×63 objective (1.4 NA) at Nyquist settings for downstream deconvolution. Images were deconvolved with Huygens Essential version 23.10 (Scientific Volume Imaging, The Netherlands, http://svi.nl) using the Huygens deconvolution wizard as a means of increasing image resolution and signal to noise. Deconvolved images were analyzed with IMARIS 7.5.2 software as described above. The fractions of postsynaptic CDH12 and CDH13 apposed to SYT2+ and CB1R+ perisomatic boutons were then quantified using the "Colocalize spots" tool (0.6 μm and 1 μm distance thresholds were applied from SYT2+ and CB1R+ boutons, respectively).

**In situ hybridization analysis.** Confocal z-stacks (1 μm step size, 1024 × 1024 resolution, 16 bits) were taken using an x63 objective (1.4 NA, 2.2 digital zoom) on an inverted Leica TCS-SP8 microscope, and maximum z-stack projections were analyzed using a custom macro in Fiji (ImageJ). Somas of infected L5 IT and L5 ET neurons were manually drawn to create a mask of the soma surface. *Cdh12* and *Cdh13* RNA particles were detected automatically based on their intensity threshold, and a mask for each probe was generated using the "Analyze Particles" tool. *Cdh12* and *Cdh13* expression levels were determined as a percentage of the probe area normalized by the soma area.

**Monosynaptic tracing analysis.** Z-stacks (5 μm step size) were taken on an inverted Zeiss ApoTome (10 X, 1280 × 800, 16 bits). For each mouse included in the analysis, 8 slices covering the entire S1 area were imaged, and infected cells were manually quantified using the Cell Counter tool in Fiji (ImageJ). The proportion of starter cells was determined as the number of receptor+/rabies+ cells normalized by the number of receptor+ cells across the 8 slices for both L5 pyramidal neuron populations. The proportion of PV + L5 IT and L5 ET inputs was determined as the number of PV+ input cells divided by the total number of input cells. The laminar proportion of PV+ input cells was determined as the number of PV+ input cells in each layer divided by the total number of PV+ input cells.

## In vitro patch-clamp recordings

Mice were deeply anesthetized with an overdose of sodium pentobarbital and transcardially perfused with 10 mL ice-cold slicing solution containing (in mM): 87 NaCl, 75 sucrose, 26 NaHCO$_3$, 11 glucose, 7 MgCl$_2$, 2.5 KCl, 1.25 NaH$_2$PO$_4$ and 0.5 CaCl$_2$, oxygenated with 95% O$_2$ and 5% CO$_2$. The brain was quickly removed, and the injected hemisphere was glued to a cutting platform before being submerged in ice-cold slicing solution. 300 μm thick coronal slices containing S1Bf were cut using a vibratome (Leica VT1200S, Wetzlar, Germany) and incubated for 45–60 min at 32 °C, and subsequently at room temperature, in the same solution. All salts were purchased from Sigma-Aldrich (St. Louis, MO). Slices were transferred to the recording setup and superfused with recording ACSF containing (in mM) 124 NaCl, 1.25 NaH$_2$PO$_4$, 3 KCl, 26 NaHCO$_3$, 10 Glucose, 2 CaCl$_2$, and 1 MgCl$_2$, which was oxygenated with 95% O$_2$ and 5% CO$_2$ and heated to 34 °C. Pipettes (3–5 MΩ) were made from borosilicate glass capillaries using a PC-10 pipette puller (P10, Narishige, London, UK). Miniature excitatory and inhibitory postsynaptic currents (mEPSCs and mIPSCs) were measured using an intracellular solution containing (in mM) 125 Cs-gluconate, 5 CsCl, 4 NaCl, 10 HEPES, 0.2 EGTA, 2 K2-phosphocreatine, 2 Mg-ATP, 0.3 GTP, adjusted with KOH to pH 7.3 (±290 mOsm). Recordings were performed at a holding voltage of −65 mV (mEPSCs) and +10 mV (mIPSCs) in the presence of 1 μM tetrodotoxin (TTX, Hello Bio, Bristol, HB1035). Recordings were made using a Multiclamp 700B amplifier (Molecular Devices, San Jose, CA). The signal was passed through a

Hum Bug Noise Eliminator (Digitimer, Welwyn Garden City, UK), sampled at 10 kHz, and filtered at 3 kHz using a Digidata 1440A (Molecular devices, San Jose, CA). Cells were excluded if the access resistance (Ra) exceeded 30 MΩ, and at least one cell per slice per mouse was included in our analysis. mEPSC and mIPSCs were analyzed using MiniAnalysis (SynaptoSoft, Decatur, GA, USA).

### Western blot
For Western blot analysis, HEK293T were rinsed with 1x ice-cold PBS. Samples were homogenized in lysis buffer containing 25 mM Tris-HCl pH 8, 50 mM NaCl, 1% Triton X-100, 0.5% sodium deoxycholate, 0.001% SDS and protease inhibitor cocktail (cOmplete Mini, Roche). Samples were resolved by SDS-PAGE and transferred onto PVDF membranes. Membranes were blocked with 5% Blotting-Grade Blocker (Bio-Rad, #1706404) in TBST (20 mM Tris-HCl pH7.5, 150 mM NaCl and 0.1% Tween20) for 1 h. After incubation with chicken anti-HA (1:10,000, Abcam Ab1190) and mouse anti-actin (1:20,000, Sigma A3854) HRP-conjugated antibodies for 1 h at room temperature, protein levels were visualized by chemiluminescence. Blots were scanned using a LI-COR Odyssey® Fc Imaging System.

### Statistical analysis
All statistical analyses were performed using GraphPad Prism 9 (GraphPad Software). Unless otherwise stated, parametric data were analyzed by two-sided unpaired and paired t-tests, one-way ANOVA followed by Holm–Sidak or Tukey's *post hoc* analysis and two-way ANOVA followed by Benjamini, Krieger and Yekutieli *post hoc* analysis for comparisons of multiple samples. Non-parametric data were analyzed by the Mann–Whitney test or Kruskal–Wallis one-way analysis followed by Holm–Sidak or Dunn's *post hoc* analysis for comparisons of multiple samples. The significance threshold was held at 0.05 (*$p < 0.05$; **$p < 0.01$; ***$p < 0.001$). All data are presented as mean ± SEM.

### Reporting summary
Further information on research design is available in the Nature Portfolio Reporting Summary linked to this article.

## Data availability
Sequencing data generated in this study have been deposited at the National Center for Biotechnology Information BioProjects Gene Expression Omnibus (GEO) and are accessible through GEO Series accession number GSE237311. Sequencing data from ref. 26 and ref. 80 were also used in this study (see Methods) and are accessible through the GEO Series accession numbers GSE122742 and GSE185862, respectively. The data used and generated in this study are openly available from King's College London research data public repository, KORDS, at https://doi.org/10.18742/28465505. The unpublished transcriptomics data can be found at https://doi.org/10.18742/28465505 (labeled as Source data_Fig2_FigS3_Klingler RNAseq). Source data are provided with this paper.

## Materials availability
All newly developed plasmids in this paper will be shared by the lead contact upon request.

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

## Acknowledgements

We are thankful to L. Doglio, F. Sanchez-Roman, S. Sanalidou and T. Garces for technical assistance and lab support, and to Ian Andrew for mouse management. We thank E. Klingler (VIB) and D. Jabaudon (University of Geneve) for kindly sharing their unpublished ET transcriptome. The pAAV *hSyn-fDIO-MCS* construct was kindly shared by M. Selten and O. Marín (King's College London). The *pSAD ΔG F3* construct was kindly provided by M. Tripodi (University of Cambridge). B7GG and HEK293-TVA cells were generously shared by T. Karayannis (University of Zurich). The *pSAD ΔG ChR2c::T2A::tdTomato* construct and the *RVdG-EnvA-tdTomato-ChR2* viral preparation were generously shared by K. Conzelmann (Ludwig-Maximilians-University Munich) and A. Delogu (King's College London). We thank the IoPPN Genomics & Biomarker Core Facility, the Advanced Cytometry Platform of the Research and Development Department at Guy's and St Thomas' NHS Foundation Trust for their technical advice and assistance on the FACS experiment, and the CRG Genomics Core Facility of Barcelona for conducting RNA sequencing. Finally, we are grateful to O. Marín, J. Burrone, M. Grubb, C. Bernard and M. Selten for critical reading of the manuscript, and to all members of the Rico and Marín laboratories for stimulating discussions and ideas. This project was supported by grants from Wellcome Trust (202758/Z/16/Z) and European Union's Horizon 2020 research and innovation program (the results leading to this publication have received funding from the Innovative Medicines Initiative 2 Joint Undertaking under grant agreement No 777394 for the project AIMS-2-TRIALS. This Joint Undertaking receives support from the European Union's Horizon 2020 research and innovation programme and EFPIA and AUTISM SPEAKS, Autistica, SFARI) to B.R., and the EMBO Long-Term Fellowship to J.J. and G.C.

## Author contributions

J.J. and B.R. conceived and designed the study. J.J. performed most of the experiments and analyzed all data. G.C. designed and produced the molecular tools, with the help of T.G. and P.M. G.C. and T.G. conducted the rabies production. T.G. and P.M. helped with the in vitro experiments and the AAV virus production. S.S., T.G., P.M. and Z.H. helped with the tissue preparation and immunohistochemistry. T.K. also contributed to collecting and analyzing data on the synaptic developmental timeline. S.S. performed the *Cdh12/13* KD in situ validation and analyzed the data. F.H. performed the RNA-Seq data analysis and visualization. T.K. and M.B. performed the ex-vivo recordings and analyzed the data. G.C. performed the *in utero* electroporations with the help of S.S., J.J. and B.R. wrote the manuscript with inputs from all authors.

## Competing interests

The authors declare no competing interests.
