## [Transparent Peer Review file · Nature Communications]

Cadherins orchestrate specific patterns of perisomatic inhibition onto distinct pyramidal cell populations

Corresponding Author: Professor Beatriz Rico

Version 0:

Reviewer comments:

Reviewer #1

(Remarks to the Author)

This manuscript by Jézéquel and co-authors addresses in mice the question of the mechanisms that control perisomatic inhibition of different types of pyramidal neurons in the cortex. The starting hypothesis is that the different patterns of innervation observed in L5 IT versus ET pyramidal neurons are generated via the expression of different adhesion molecules for different pairs of input/target neurons. The authors 1) describe, using immunolabeling and patch-clamp recordings, the development of the inhibitory innervation of L5 IT and ET neurons, 2) perform comparative transcriptomics, 3) test the involvement of two cadherins, and 4) map inhibitory connectivity on L5 neurons using tracing technologies.

Overall, the results show that:

- 1) L5 ET neurons receive more PV inhibition than L5 IT neurons, as has been shown in other cortical areas. This correlates with higher expression of cadherin-13 in ET Neurons and knockdown of cadherin-13 in ET Neurons seems to reduce the amount of PV inhibition.
- 2) A majority of L5 IT Neurons receive an additional, albeit minor, fraction of inhibitory input from CCK cells. Knockdown of cadherin-12 seems to reduce the amount of CCK innervation of IT neurons.

The topic of this study is of high interest and the data presented by the authors bring useful information to the community. However, I feel that the data as presented in the manuscript cannot support the conclusions made by the authors, that is that “the specific innervation by inhibitory inputs on different types of pyramidal neurons is controlled by differences in the transcriptional programs of the target neurons”. An alternative interpretation of their data could be that the location of the pyramidal neurons in the different L5 sublayers is the key parameter that drives the different patterns of innervation by inhibitory inputs. Because there are more IT neurons in L5a, the authors observe more innervation by CCK interneurons, which do not seem to reach much beyond this layer.

To exclude this possibility, the authors would need to analyze their data according to the position of the target neurons in L5a versus L5b (or only compare data from ET and IT neurons localized in L5b). While the authors have made an effort in this direction, further analysis is needed to support the conclusions of the manuscript. I also think that some critical controls are missing in the knockdown experiments, as described below.

1) The data presented in the figure for reviewer 1 point 1 show the densities of PV inhibitory boutons according to the location of IT neurons (and should be added to the manuscript). They confirm that IT neurons receive less PV inhibition than ET neurons, regardless of their position in the layer. However, I would have expected to have the same analysis for CCK boutons. In sup fig1c, what does the distribution look like if the % of CCK inputs is plotted separately as a function of the position in L5a or L5b?

The image in figure 1b for reviewers is insufficient, as it is difficult to assess the staining at this magnification. But it does suggest that indeed the CCK innervation of L5 neurons is only a question of their location in different sublayers, not of neuronal transcriptional identity.

The data presented in Fig 1c and 1d for reviewers would have been more helpful if the L5 IT results were divided according to their location in sublayers. As it is, they only confirm that the majority of ET neurons receive no CCK innervation at all. Also, because they are presented as mean per animal, these data could hide very different distributions. What is the

magnitude of CCK inhibition received by ET neurons when they receive one ? Is it really significant functionally? Furthermore, the data are extremely variable for IT neurons: is this because of differences in sampling between L5a and L5b?

2) In their response to reviewer 1, the authors are right to say that their mean per animal takes in account intra and inter variability, but these averages could hide very different distributions. And of course the authors are right to say that taking the cells as the n for statistical comparison is not correct (they should not use it in the figures of the article, as for ex insets in sup fig3). However, statistical analysis (using linear mixed models, see for example <https://stats.oarc.ucla.edu/other/mult-pkg/introduction-to-linear-mixed-models/>) can now help tease apart the effects in datasets that are not independent (that is, when you have several measurements per animal), while still presenting the variability between cells.

While the different tests are described in the statistical analysis section, the rationale behind the choice of the tests is not explained. For example, in main text figure 1e, h why not use a two-way ANOVA (developmental time point x neuron type)?

3) In fig.2 the authors select two cadherins as potential cell surface molecules controlling the differences in perisomatic inhibition of different types of L5 neurons. They conclude that the differences in expression of Cadherin-12 and Cadherin-13 mirror cell-type specific PV+ and CCK+ connectivity motifs. The distributions shown in sup figs 3d and 3g are very helpful and they actually show quite some variability in the level of expression of the two cadherins by each population of L5 pyramidal neurons. Again it would have been interesting to show whether this variability relates to the position in the sublayers.

The distribution of tagged cadherins in sup fig4 is quite difficult to visualize and the results of the quantification are difficult to interpret. Because it is overexpression (I do understand the difficulty of finding antibodies for immunolocalization), I am not sure one can say much besides the fact that both cadherins, to a little extent, can be targeted close to both types of inhibitory synapses.

5) I have some concerns about the knockdown strategy. The authors need to be explicit in the text about the levels of KD they achieve with their strategy (line 183). These levels are low and this will affect the interpretation of the data. Additional controls are necessary, and would strengthen the interpretation of the results. There is no control of off-target effects, besides the fact that they show the specificity in reducing cdh12 versus cdh13 expression. A typical control when using shRNA is to co-express rescue constructs (see for example Favuzzi et al. 2019 by the same group). The results show a higher magnitude of KD for Cdh13 in IT neurons compared to ET neurons. Yet, they only see a difference in PV innervation of ET Neurons after Cdh13 KD. That would suggest that Cdh13 has no role at all in PV Innervation of IT neurons while it is expressed in these neurons (sup fig 3). Similarly the magnitude of the effect of Cdh12 KD on CCK synapse densities in IT neurons is much higher than expected from the levels of KD. How do the authors explain this ?

6) The retrograde mapping confirms a layer effect: more superficial PV neurons connect to more superficial pyramidal neurons. I am not sure it brings more than this and I am not sure I understand the logic of the non-input versus input analysis of Cdh13 expression levels. Not all ET Neurons are labelled using their strategy, so I suppose it is possible that the non-input interneurons can still target ET cells?

Minor point:

1) The authors state that the differences in perisomatic inhibition appear during the initial steps of synaptogenesis. But while this might be true for the amount of PV innervation for ET neurons, how relevant biologically are the differences in CCK innervation before P30 (I mean to what number of presynaptic boutons per cell does that correspond ?)

2) I find the diagrams in fig 2e and 2f misleading: L5 IT neurons receive also the PV component of inhibition

3) "% expression" in the FISH experiment is unclear.

4) At what age did they analyze HA tagged cadherins localization ?

5) In related Sup fig 1c: I do not understand what they represent. Is it % of PV inputs/total number of inputs per soma ?

Reviewer #2

(Remarks to the Author)

The authors have addressed carefully all my comments and significantly improved the manuscript. They added new slice electrophysiology data, additional FISH experiments, cell death analysis and extensive quantification. They also edited the results and discussion addressing my previous concerns. I consider this manuscript ready for publication.

Reviewer #4

(Remarks to the Author)

Jezequel et al. show a novel mechanism for cell-type specific inhibition onto layer 5 neurons. The authors have addressed the majority of my concerns and made revisions where necessary. The authors used appropriate statistical measures and reported all relevant values. I have no further issues.

Version 1:

Reviewer comments:

Reviewer #1

(Remarks to the Author)

Provided the following revisions, the manuscript by Jezequel et al. can be published:

1. Indeed the authors understood properly: as indicated in the link I provided and in the reference Arts et al. 2014, the issue is to account for non independence in the data. A nested t test can be the solution if they compare only two groups, with multiple measurements per animal.

In supplementary figure 1b, d, it seems that the n used for the statistical analysis is the number of cells, and not the number of animals. This should be corrected.

2. Characterising a shRNA strategy implies both testing its efficiency and specificity and thus, both experiments were expected after the initial review. If the authors cannot perform the rescue experiments due to the size of the cDNA, this limitation should be acknowledged in the manuscript. In the same way that the authors state that L5 IT and ET neurons express different interactomes to explain the cell type- specific impact of the knockdowns, one could argue that the potential off-targets are not expressed in both neuron types at the same level. However I agree that the fact that each shRNA leads to knockdown of expression for one cadherin and not the other gives some level of confidence on the specificity of the strategy.

3. In figure 5d and e, the results should be divided for L5 IT in L5a and L5b as in sup figure 1. This would help confirm the interpretation of the data made by the authors (that it is not a question of more superficial PV neurons connecting to more superficial pyramidal neurons).

Response to reviewer's comments on NCOMMS-24-04426-T_R1

We thank the reviewers for their thoughtful comments and criticisms, which were overall positive and constructive and helped strengthen the manuscript. While Reviewers 2 and 4 did not raise any further issues, Reviewer 1 made several additional suggestions. We have addressed all of them, as described below. In this document, original comments made by the reviewers are in blue, and our responses are in black.

Reviewer #1:

Although the reviewer acknowledges the high interest of our study, they still have concerns about our interpretation of the results and suggest an alternative interpretation. The main discussion here is whether molecular mechanisms underlying cell-specific synaptic targeting are driven by cell type identity (e.g. ET versus IT) or the laminar position of pyramidal neurons. We have addressed all the reviewer's concerns in our detailed responses below.

1) The data presented in the figure for reviewer 1 point 1 show the densities of PV inhibitory boutons according to the location of IT neurons (and should be added to the manuscript). They confirm that IT neurons receive less PV inhibition than ET neurons, regardless of their position in the layer. However, I would have expected to have the same analysis for CCK boutons. In sup fig1c, what does the distribution look like if the % of CCK inputs is plotted separately as a function of the position in L5a or L5b? The image in figure 1b for reviewers is insufficient, as it is difficult to assess the staining at this magnification. But it does suggest that indeed the CCK innervation of L5 neurons is only a question of their location in different sublayers, not of neuronal transcriptional identity.

We apologize if the image provided in **Figure 1b for reviewer** misled Reviewer 1's interpretation. In the previous revision, we already showed that L5 IT neurons in L5a and in L5b receive similar levels of PV+ inhibition (**same cell type in different sub-layers receive similar innervation levels**). However, L5 ET neurons also present in L5b receive a higher density of PV+ inputs than L5b IT neurons (**different cell types in the same sub-layer receive different innervation levels**). Following the reviewer's request, we have also quantified CCK+ input densities onto L5 IT and L5 ET pyramidal neurons according to their sublayer position. As expected, we observed a gradual decrease of CCK+ innervation from L5a to deep L5b. L5 IT neurons located in L5a showed higher levels of CCK+ inhibition compared to L5 IT neurons in L5b. However, the density of CCK+ inputs targeting L5b IT neurons was significantly higher than onto neighbouring L5 ET neurons. Altogether, our results demonstrate that neighbouring cells with different identities do not share the same inhibition patterns. We hope these additional results will convince the reviewer that cell-specific inhibition patterns do not simply result from the target cell's location but are instructed by the target cell identity and cell-type specific transcriptional programs.

Actions taken:

- We have compared CCK+ basket cell input densities onto L5 IT neurons in L5a and L5b with L5 ET neurons (in L5b).
- We also provide updated PV/CCK input ratios for each subpopulation (previously Suppl. Figure 1c).
- Both PV+ and CCK+ datasets have been included in Supplementary Figure 1.

The data presented in Fig 1c and 1d for reviewers would have been more helpful if the L5 IT results were divided according to their location in sublayers. As it is, they only confirm that the majority of ET neurons receive no CCK innervation at all. Also, because they are presented as mean per animal, these data could hide very different distributions. What is the magnitude of CCK inhibition received by ET neurons when they receive one? Is it really significant functionally?

As our manuscript clearly states, CCK+ inhibition remains largely outnumbered by PV+ inhibition in both populations. Therefore, it is challenging to disentangle the contribution of CCK+ innervation to whole pyramidal cell inhibition. This contribution is likely not about pure inhibition but more modulatory effects on the entire pyramidal cell function (please see the Discussion section).

Furthermore, the data are extremely variable for IT neurons: is this because of differences in sampling between L5a and L5b?

L5 IT neurons were randomly imaged in L5a and L5b in order to minimize sampling differences between sublayers. We also observed some dispersion in L5 ET neurons (Figure 1d), a very homogenous cell population compared to L5 IT. Data dispersion is likely intrinsic to the experimental design (different mice, litters, etc.).

2) In their response to reviewer 1, the authors are right to say that their mean per animal takes in account intra and inter variability, but these averages could hide very different distributions. And of course the authors are right to say that taking the cells as the n for statistical comparison is not correct (they should not use it in the figures of the article, as for ex insets in sup fig3). However, statistical analysis (using linear mixed models, see for example <https://stats.oarc.ucla.edu/other/mult-pkg/introduction-to-linear-mixed-models/>) can now help tease apart the effects in datasets that are not independent (that is, when you have several measurements per animal), while still presenting the variability between cells. While the different tests are described in the statistical analysis section, the rationale behind the choice of the tests is not explained. For example, in main text figure 1e, h why not use a two-way ANOVA (developmental time point x neuron type)?

We thank the reviewer for their input on statistical analysis, and will address their comment that raises two independent points: 1) the use of linear mixed models instead of averaging data per animal, and 2) the rationale behind the statistical tests performed in Figure 1e and h.

We decided to use a multiple t-test to assess whether PV+ (Figure 1e) or CCK+ (Figure 1h) inhibition differed between L5 IT and L5 ET populations (paired data – two cell types labelled in the same brain) at specific time points, each time point representing an independent dataset. We agree with the reviewer that a two-way ANOVA is best suited and thank them for this helpful suggestion. We have now included this analysis in the revised version of the manuscript (Figure 1e, h).

Regarding the first point, the reviewer advises using linear mixed models that can help “tease apart the effects in datasets”. Linear mixed models are indeed useful in cases where nesting of the data can mask true effects. The second image in the resource linked to by the reviewer shows that the data's nesting masks the predictor's effect on the outcome. In other words, there is a relationship between predictor and outcome within each group at the higher level. However, this is not the case in our datasets. Since every cell in each animal belongs to the same categorical group (e.g. control vs shRNA), no relationship between predictor and outcome can be measured within an individual animal. We are, therefore, unsure what effects the reviewer is referring to that can be teased apart. We assume they might refer to a nested analysis rather than a linear mixed model (see Aarts et al., 2014).

As our previous answer to Reviewer 1 explained, using the means per animal is a valid statistical method. We are nevertheless open to discussing the need to perform additional analysis if the reviewer and the editor think it is critical for the manuscript.

3) In fig.2 the authors select two cadherins as potential cell surface molecules controlling the differences in perisomatic inhibition of different types of L5 neurons. They conclude that the differences in expression of Cadherin-12 and Cadherin-13 mirror cell-type specific PV+ and CCK+ connectivity motifs. The distributions shown in sup figs 3d and 3g are very helpful and they actually show quite some variability in the level of expression of the two cadherins by each population of L5 pyramidal neurons. Again it would have been interesting to show whether this variability relates to the position in the sublayers.

We are glad the reviewer finds the distributions helpful. To address the reviewer's point, we have segregated *Cdh12* and *Cdh13* expression in L5 IT and L5 ET neurons according to the position of pyramidal neurons in L5a and L5b sublayers. Our results show that *Cdh12* expression is similar in L5 IT neurons located in L5a and L5b and significantly higher than in L5 ET neurons. Similarly, *Cdh13* expression is significantly higher in L5 ET neurons than in L5 IT neurons, independently of their location in L5. These results further confirm that *Cdh12* and *Cdh13* expression is cell-type specific rather than layer-specific.

Actions taken:

- We have included this new data in Supplementary Figures 3d and 3g.

The distribution of tagged cadherins in sup fig4 is quite difficult to visualize and the results of the quantification are difficult to interpret. Because it is overexpression (I do understand the difficulty of finding antibodies for immunolocalization), I am not sure one can say much besides the fact that both cadherins, to a little extent, can be targeted close to both types of inhibitory synapses.

As we stated in the manuscript, co-labeling of PV+ and CCK+ boutons revealed that postsynaptic CDH13-HA preferentially clustered near the postsynaptic sites of PV+ boutons, while CDH12-HA did not show any preferential localization near PV+ or CCK+ synapses.

*5) I have some concerns about the knockdown strategy. The authors need to be explicit in the text about the levels of KD they achieve with their strategy (line 183). These levels are low and this will affect the interpretation of the data. Additional controls are necessary, and would strengthen the interpretation of the results. There is no control of off-target effects, besides the fact that they show the specificity in reducing *cdh12* versus *cdh13* expression. A typical control when using shRNA is to co-express rescue constructs (see for example Favuzzi et al. 2019 by the same group). The results show a higher magnitude of KD for *Cdh13* in IT neurons compared to ET neurons. Yet, they only see a difference in PV innervation of ET Neurons after *Cdh13* KD. That would suggest that *Cadh13* has no role at all in PV Innervation of IT neurons while it is expressed in these neurons (sup fig 3). Similarly the magnitude of the effect of *Cdh12* KD on CCK synapse densities in IT neurons is much higher than expected from the levels of KD. How do the authors explain this ?*

In the previous revision, Reviewer 1 suggested that the shRNA strategy should be better characterised and asked to perform additional experiments to check if the level of knockdown was the same regardless of the L5 population targeted. The efficiency of our shRNA strategy ranged from around 20% (*Cdh13* in ET), 40% (*Cdh13* in IT), 30% (*Cdh12* in IT) and 20%

(*Cdh12* in ET). These numbers are now stated in the main text as requested by the reviewer, and are consistent with published work from our lab (Favuzzi et al., Neuron 2017; Favuzzi et al., Science 2019; Bernard et al., Science 2022). It is, however, difficult to draw direct parallels between the efficacy of the knockdown at the RNA level and the magnitude of the synaptic phenotype.

We report that a 22% reduction of *Cdh13* expression in L5 ET neurons was sufficient to reduce PV inhibition onto L5 ET neurons. However, a 36% *Cdh13* knockdown in L5 IT neurons did not lead to any synaptic phenotype. On the other hand, similar *Cdh12* down-regulation levels in both L5 populations only caused a CCK+ synaptic phenotype in L5 IT neurons. We reported in several of our studies that when a gene is relevant for synaptic function, a mild knockdown is sufficient to trigger synaptic phenotypes. Conversely, very strong knockdowns (e.g. 80%) can also fail to lead to any synaptic phenotype (Kroon et al., BioRxiv 2024). Our results thus clearly highlight the cell-type-specific impact of *Cdh12* and *Cdh13* shRNA down-regulation. As stated in the discussion, it is conceivable that L5 IT and L5 ET neurons express different interactomes and may not contain the relevant postsynaptic machinery to allow CDH13 and CDH12 to instruct PV+ and CCK+ basket cell wiring, respectively.

For this second revision, the reviewer now requests a rescue of the shRNA synaptic effect following a strategy we previously used (Favuzzi et al., Science 2019), as an additional control for potential off-target effects. In contrast to the genes explored in Favuzzi et al (Science 2019), *Cdh12* (2379) and *Cdh13* (2184) are unfortunately too large to be packed with the shRNA constructs in an AAV. Performing such a rescue experiment would require designing a new construct for *in-utero* electroporation, combined with an inducible temporal Cre to drive post-natal recombination (electroporation + retroAAV injection). Even if this challenging experiment would work, it would extend the revision to an extra 9 months to a year at best. Furthermore, we showed that *Cdh12* or *Cdh13* down-regulation does not affect the levels of the other Cadherin (Supplementary Figure 6). We believe that if a molecule part of the same family with similar sequences is not affected by non-specific down-regulation, the chances for off-targets are very low. In addition, we cannot think of any scenario where a shRNA leads to off-targets, causing a synaptic phenotype in one specific pyramidal cell and not the other (e.g. *shCdh13* leading to a PV synaptic phenotype in ET and not IT cells, Figure 3 and Supp. Fig. 6). We hope the reviewer understands that such evidence supports our strategy's specificity, in addition to all our published studies.

6) The retrograde mapping confirms a layer effect: more superficial PV neurons connect to more superficial pyramidal neurons. I am not sure it brings more than this and I am not sure I understand the logic of the non-input versus input analysis of Cdh13 expression levels. Not all ET Neurons are labelled using their strategy, so I suppose it is possible that the non-input interneurons can still target ET cells?

We are unsure what the reviewer means by “the retrograde mapping confirms a layer effect”. The layer effect previously mentioned by the reviewer referred to the position of pyramidal neurons within L5 (L5a versus L5b). Our retrograde mapping experiment here shows that 1) PV+ neurons connected to L5 IT and L5 ET exhibit different laminar distributions, suggesting that different PV+ populations innervate different L5 pyramidal cell populations, 2) that L5 IT neurons located in L5a and L5b both preferentially receive connections from PV+ interneurons located in L2/3, L4 and L5a, and 3) that L5 ET cells (located in 5b) receive connections from PV+ interneurons located in L5a and L5b. Thus, we respectfully disagree with the conclusion of Reviewer 1, stating, “more superficial PV neurons connect to more superficial Pyramidal cell neurons”.

As rightly pointed out by the reviewer, rabies do not label all L5 ET neurons. Hence, a fraction of non-input PV+ cells could represent a pool of unlabelled PV+ cells that are actually connected to the L5 ET population. Therefore, if any, we are underestimating our phenotype;

in other words, we include cells expressing high levels of *Cdh13* in the pool of non-input cells that belong to the connected population. Despite this, “non-connected” cells still express less *Cdh13*.

While going through this revision, we noticed a mistake in the images selected in Fig. 5g. We have changed PV- non-inputs to PV+ non-inputs illustrations, as they are the correct representation of the quantification (Fig. 5h).

Minor point:

1) The authors state that the differences in perisomatic inhibition appear during the initial steps of synaptogenesis. But while this might be true for the amount of PV innervation for ET neurons, how relevant biologically are the differences in CCK innervation before P30 (I mean to what number of presynaptic boutons per cell does that correspond ?)

We acknowledge that it is difficult to predict the functional relevance of CCK+ inputs early in development since CCK+ innervation in L5 remains low until its dramatic increase between P15 and P30. However, our intention in exploring the emergence of CCK+ basket innervation early in development was not to investigate the functional role of CCK+ inhibition *per se* but was rather to identify a developmental time window during which different molecular programs might emerge to support differential connectivity patterns.

2) I find the diagrams in fig 2e and 2f misleading: L5 IT neurons receive also the PV component of inhibition

We have added a sentence in the legend to clarify this point.

3) “% expression” in the FISH experiment is unclear.

“% expression” corresponds to the area labelled by the probe over the soma area, as indicated in the methods section.

4) At what age did they analyze HA tagged cadherins localization ?

The localization of HA-tagged cadherins was analysed between P18-P21. This information now appears in Supplementary Figure 4.

5) In related Sup fig 1c: I do not understand what they represent. Is it % of PV inputs/total number of inputs per soma?

We apologise if this was unclear. The reviewer is correct; it indeed represents the percentage of PV or CCK inputs over the total number of perisomatic inputs. We have relabelled the graph axis of Supplementary Figure 1c and its legend accordingly.

Response to reviewer's comments on NCOMMS-24-04426A_R3

We thank the reviewer for their final comments, which we have addressed below. In this document, original comments made by the reviewers are in blue, and our responses are in black.

Reviewer #1:

Provided the following revisions, the manuscript by Jezequel et al. can be published:

1. Indeed the authors understood properly: as indicated in the link I provided and in the reference Arts et al. 2014, the issue is to account for non independence in the data. A nested t test can be the solution if they compare only two groups, with multiple measurements per animal. In supplementary figure 1b, d, it seems that the n used for the statistical analysis is the number of cells, and not the number of animals. This should be corrected.

The revised version of this manuscript incorporates the corrections requested by the reviewer for Supplementary Figure 1. As mentioned in our previous revision and emphasised by Reviewer 4, we utilised appropriate and valid statistical methods and would prefer not to alter the entire statistical design.

2. Characterising a shRNA strategy implies both testing its efficiency and specificity and thus, both experiments were expected after the initial review. If the authors cannot perform the rescue experiments due to the size of the cDNA, this limitation should be acknowledged in the manuscript. In the same way that the authors state that L5 IT and ET neurons express different interactomes to explain the cell type- specific impact of the knockdowns, one could

argue that the potential off-targets are not expressed in both neuron types at the same level. However I agree, that the fact that each shRNA leads to knockdown of expression for one cadherin and not the other gives some level of confidence on the specificity of the strategy.

We have added a sentence in the Methods to acknowledge the technical limitations preventing us from performing a rescue experiment.

3. In figure 5d and e, the results should be divided for L5 IT in L5a and L5b as in sup figure 1. This would help confirm the interpretation of the data made by the authors (that it is not a question of more superficial PV neurons connecting to more superficial pyramidal neurons).

The results in Figure 5e provide a more detailed analysis of the data presented in Figure 5d, illustrating the distribution of PV+ inputs in L5a and L5b for the L5 IT and ET populations. Once again, the data from our rabies experiments support our conclusions: L5 ET neurons located in Layer 5b receive PV inputs from both layers, L5a and L5b. Unfortunately, we cannot differentiate PV+ inputs associated with L5a IT or L5b IT starter cells using this multiplex monosynaptic tracing approach. Inputs from L5a and L5b IT starter cells could potentially be distinguished by 1) conducting single-cell monosynaptic tracing or 2) utilizing genetic mouse lines that would enable sublayer targeting of L5 IT (L5a IT and L5b IT) pyramidal neurons, of which we are currently unaware. We hope the reviewer appreciates that we are unable to fulfil their request due to the reasons stated above.

Kind regards,

Beatriz Rico